# Enhancing photocatalytic $H_2O_2$ production with Au co-catalysts through electronic structure modification

Xidong Zhang [1], Duoduo Gao [2], Bicheng Zhu [1], Bei Cheng[3], Jiaguo Yu [1] & Huogen Yu [1] ✉

Gold-based co-catalysts are a promising class of materials with potential applications in photocatalytic $H_2O_2$ production. However, current approaches with Au co-catalysts show limited $H_2O_2$ production due to intrinsically weak $O_2$ adsorption at the Au site. We report an approach to strengthen $O_2$ adsorption at Au sites, and to improve $H_2O_2$ production, through the formation of electron-deficient $Au^{\delta+}$ sites by modifying the electronic structure. In this case, we report the synthesis of $TiO_2/MoS_x$-Au, following selective deposition of Au onto a $MoS_x$ surface which is then further anchored onto $TiO_2$. We further show that the catalyst achieves a significantly increased $H_2O_2$ production rate of 30.44 mmol $g^{-1}$ $h^{-1}$ in $O_2$-saturated solution containing ethanol. Density functional theory calculations and X-ray photoelectron spectroscopy analysis reveal that the $MoS_x$ mediator induces the formation of electron-deficient $Au^{\delta+}$ sites thereby decreasing the antibonding-orbital occupancy of Au-$O_{ads}$ and subsequently enhancing $O_2$ adsorption. This strategy may be useful for rationally designing the electronic structure of catalyst surfaces to facilitate artificial photosynthesis.

Solar-driven photocatalytic $H_2O_2$ production through oxygen reduction reaction (ORR) is a promising method for addressing energy and environmental crises due to its low energy consumption, safety, and environmental friendliness[1–5]. However, the production of $H_2O_2$ via photocatalysts is hindered by the low efficiency of electron transfer and interface reaction[6–8], resulting in suboptimal $H_2O_2$ yields[9,10]. To address these challenges, cocatalyst deposition on the surface of photocatalysts can not only effectively promote electron transfer but also provide specialized active sites to facilitate interfacial ORR[11–13]. It is well known that photocatalytic $H_2O_2$ production via ORR on the active sites of cocatalysts involves multiple fundamental steps, such as $O_2$ adsorption, intermediate *OOH formation, and $H_2O_2$ desorption[8,14,15]. Of these, $O_2$ adsorption at active sites is one of the most important processes, as it facilitates the formation of intermediate *OOH and its further conversion into $H_2O_2$[16–19]. Sabatier principle[20]. Suggests that the interaction between active sites and adsorbates must have an optimal binding energy. Further research indicates that the electron configuration of active sites fundamentally determines their interaction with adsorbates, influencing their adsorption/desorption performances[21]. However, in photocatalytic $H_2O_2$ production, current cocatalysts usually suffer from a mismatch in the electronic configuration between the active site and the adsorbed $O_2$, leading to either excessively strong or weak $O_2$ adsorption, which in turn limits $H_2O_2$-production rates[22–24]. Therefore, it is quite meaningful and challenging to modulate the electronic configuration of cocatalysts and optimize their oxygen adsorption strength to achieve efficient $H_2O_2$ production.

Currently, noble metal cocatalysts (Pd, Pt, and Au) have made significant advancements in improving photocatalytic ORR for $H_2O_2$ production[25–28]. Notably, Au cocatalysts usually exhibit a higher photocatalytic $H_2O_2$-production activity, attributed to the effective

[1]Laboratory of Solar Fuel, Faculty of Materials Science and Chemistry, China University of Geosciences, 68 Jincheng Street, Wuhan, P. R. China. [2]State Key Laboratory of Silicate Materials for Architectures, Wuhan University of Technology, 122 Luoshi Road, Wuhan, P. R. China. [3]State Key Laboratory of Advanced Technology for Material Synthesis and Processing, Wuhan University of Technology, Wuhan, P. R. China. ✉e-mail: yuhuogen@cug.edu.cn

interfacial charge transfer between photocatalysts and Au nanoparticles[29]. This transfer enables the rapid movement of photo-generated electrons from the photocatalysts to the Au surface, facilitating the reduction of adsorbed $O_2$ to $H_2O_2$ through either a two-step single-electron or a one-step two-electron ORR process[30–32]. However, metallic Au usually exhibits weak oxygen adsorption characteristics due to its intrinsic electronic structure (Fig. 1a-(1))[33,34], which limits the formation of the *OOH intermediate and subsequent $H_2O_2$ production. Consequently, precise modulation of Au's electronic structure is extremely crucial to optimize $O_2$ adsorption and enhance photocatalytic $H_2O_2$ production. For instance, Tsukamoto et al. demonstrated increased electronic density in Au by forming an Au-Ag alloy, reducing $H_2O_2$ decomposition on Au sites[35]. Moreover, Wang et al. prepared an efficient core-shell Cu@Au-modified $BiVO_4$ nanostructure. This structure reduces negative charge accumulation at the Au active sites by forming an ohmic contact with $Cu/BiVO_4$, thereby enhancing the adsorption of $O_2$ and its intermediate *OOH, leading to efficient photocatalytic $H_2O_2$-production performance[36]. Although the introduction of alloy and bimetallic core-shell structures have efficiently enhanced the photocatalytic activity of Au sites, the relationship about the $H_2O_2$-production activity, the Au-$O_{ads}$ bonds, and the electronic structure of Au remains unclear. Fortunately, the molecular orbital theory clearly states that the antibonding-orbital occupancy degree between a metal and its adsorbate usually determines its adsorption energy[37], which provides a theoretical basis for the modulation of bond strength between Au and $O_2$. Inspired by this,

selectively decreasing the antibonding-orbital occupancy of Au-$O_{ads}$ is expected to further enhance the $O_2$ adsorption on Au, potentially achieving efficient photocatalytic $H_2O_2$ production. However, there has been limited research focusing on this approach to date.

In this work, we propose an approach to strengthen the Au-$O_{ads}$ bonds by modifying the electronic structure of Au active site. This is achieved through the introduction of molybdenum sulfide ($MoS_x$) as an electron mediator to decrease the antibonding-orbital occupancy of Au-$O_{ads}$. In this case, the $MoS_x$ mediator serves to adjust the electronic structure of Au cocatalyst, resulting in the creation of electron-deficient $Au^{\delta+}$ active sites and subsequently accelerating $H_2O_2$ production (Fig. 1a-(2) and -(3)). To this end, $TiO_2/MoS_x$-Au photocatalyst was synthesized by a two-step method. This process involves the initial $MoS_x$ deposition onto the $TiO_2$ surface and subsequent S-induced selective photodeposition of Au cocatalyst onto the $MoS_x$ surface. The resulting $TiO_2/MoS_x$-Au photocatalyst achieves a boosted $H_2O_2$-production rate of 30.44 mmol g$^{-1}$ h$^{-1}$, which is 25.4 and 1.3 times higher than $TiO_2$ and $TiO_2$/Au, respectively. Density functional theory (DFT) calculations and ex-situ X-ray photoelectron spectroscopy (XPS) analysis have confirmed the effective reduction of $d$-orbital electron on Au cocatalyst upon the introduction of $MoS_x$, leading to a decrease in antibonding-orbital occupancy in Au-$O_{ads}$ (Fig. 1b). Consequently, the Au-$O_{ads}$ bonds are significantly reinforced, which in turn enhances the rate of photocatalytic $H_2O_2$ production. This work focuses on modifying the electron structure of Au cocatalyst to reduce the antibonding-orbital occupancy of Au-$O_{ads}$, offering a promising

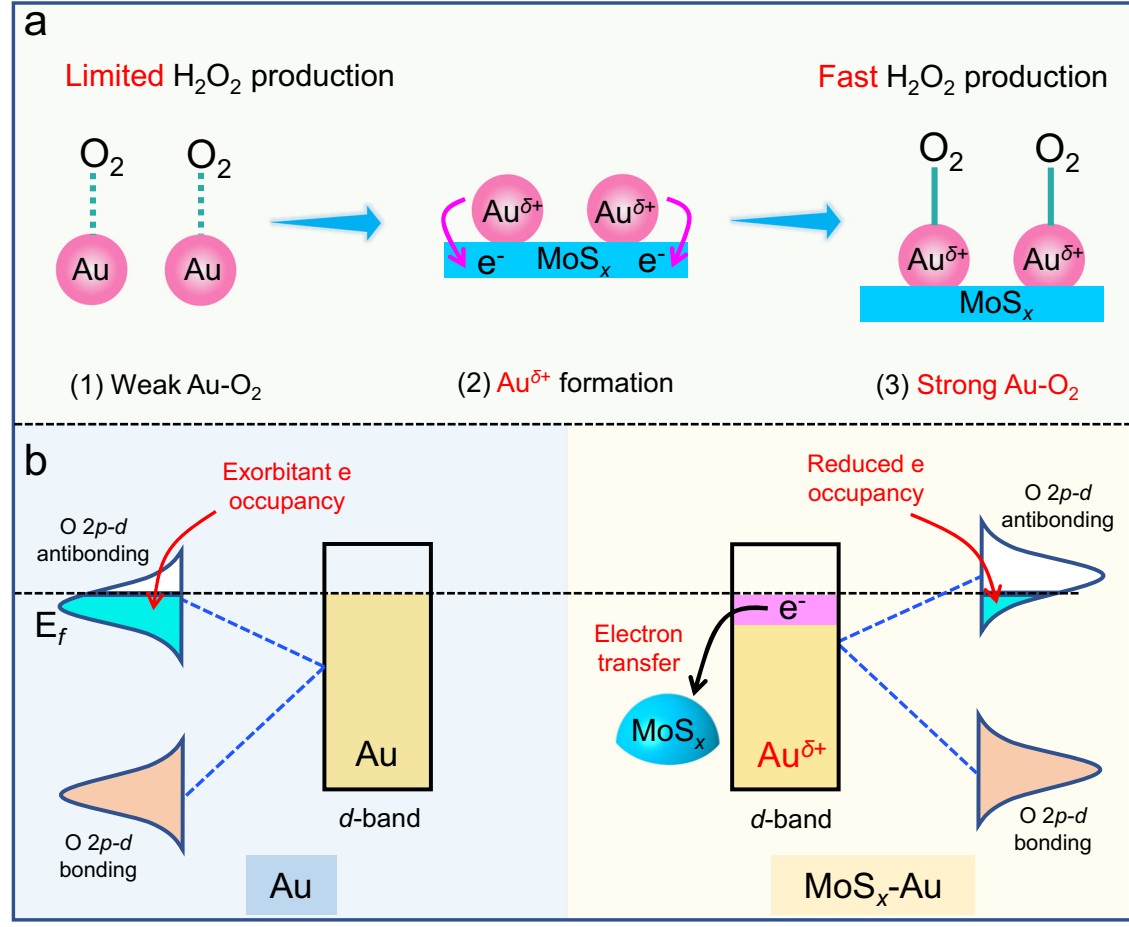

**Fig. 1 | Strategy to design efficient electron-deficient $Au^{\delta+}$ cocatalyst for improving $H_2O_2$-production kinetics. a** Schematic illustration of electron-deficient $Au^{\delta+}$ formation to reinforce Au-$O_{ads}$ bond: (1) weak Au-$O_{ads}$ bond on Au surface; (2) the formation of electron-deficient $Au^{\delta+}$ sites by $MoS_x$ incorporation; (3) strong Au-$O_{ads}$ bond on $MoS_x$-Au surface. **b** Schematic diagram about reducing the antibonding-orbital occupancy of Au-$O_{ads}$ by the free-electron transfer from Au to $MoS_x$ cocatalyst.

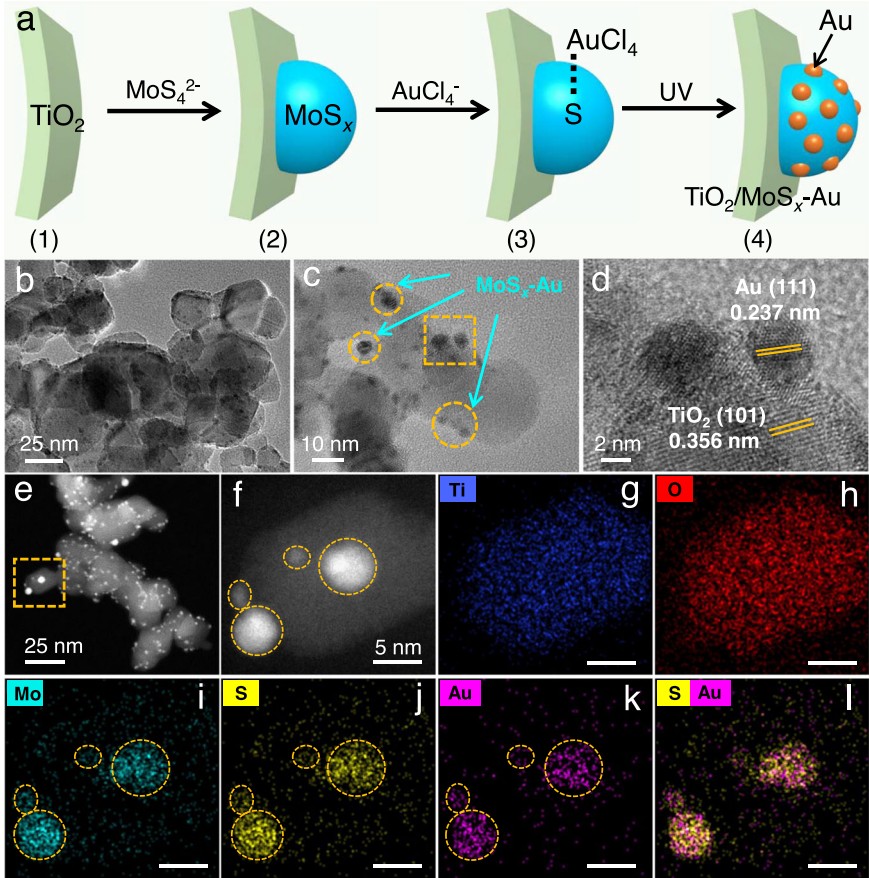

**Fig. 2 | Synthetic strategy and morphology characterization. a** Schematic illustration for the synthesis of TiO$_2$/MoS$_x$-Au by the initial lactic acid-induced MoS$_x$ deposition on the TiO$_2$ surface and subsequent S-induced selective photodeposition of Au cocatalyst onto the MoS$_x$ surface. **b, c** TEM, **d** HRTEM, **e, f** HAADF-STEM, and **g–l** elemental mapping pictures of TiO$_2$/MoS$_x$-Au photocatalyst.

approach to enhance O$_2$ adsorption for efficient photocatalytic H$_2$O$_2$ production.

## Results and discussion

### Synthesis and characterization of TiO$_2$/MoS$_x$-Au

To realize the successful deposition of MoS$_x$-Au cocatalyst on the surface of TiO$_2$ photocatalyst, a facile two-step route was carried out at room temperature (Fig. 2a), including the initial deposition of MoS$_x$ on the TiO$_2$ surface and the subsequently selective photodeposition of Au onto the MoS$_x$ surface (Supplementary Fig. 1). First, (NH$_4$)$_2$MoS$_4$ solution was mixed with a lactic acid solution to form brown MoS$_x$ colloidal nanoparticles (Supplementary Fig. 1a-(1)). Subsequently, TiO$_2$ nanoparticles were uniformly dispersed into this colloidal solution with constant stirring. The positive charge on the TiO$_2$ nanoparticles in the lactic acid solution allowed for the efficient adsorption of MoS$_x$ colloidal nanoparticles onto the TiO$_2$ surface via electrostatic self-assembly (Supplementary Fig. 1a-(2), c). The deposition of MoS$_x$ on the TiO$_2$ surface can be verified by Fourier transform infrared (FTIR) spectroscopy and Raman spectra analyses (Supplementary Fig. 2). A new FTIR peak for S-S vibration and a new Raman peak for Mo-S can be observed, providing strong evidence for the MoS$_x$ formation[38,39]. With the further addition of HAuCl$_4$ solution into the TiO$_2$/MoS$_x$ suspension, AuCl$_4^-$ ions can be selectively adsorbed onto the MoS$_x$ surface via the strong interaction between S and Au atoms (Fig. 2a)[40]. Upon light irradiation, the AuCl$_4^-$ ions were in situ reduced to form Au nanoparticles on the MoS$_x$ surface (Supplementary Figs. 1a-(3), d), evident from a color change from light brown to purple (Supplementary Fig. 1b), revealing the successful synthesis of TiO$_2$/MoS$_x$-Au

photocatalyst. The above result can further be supported by X-ray diffraction (XRD) and Raman spectra. Compared with the TiO$_2$/MoS$_x$ sample, a new XRD peak of Au at 38.1° and an Au-S Raman peak[41,42] confirm the selective deposition of Au nanoparticles on the MoS$_x$ surface (Supplementary Figs. 3 and 4).

Transmission electron microscopy (TEM) analysis was employed to further verify the selective deposition of Au on the MoS$_x$ surface within the TiO$_2$/MoS$_x$-Au photocatalyst. As depicted in Fig. 2b–d, numerous dark spots on the TiO$_2$ surface are observed, which can be attributed to the MoS$_x$-Au cocatalyst. The high-angle annular dark-field (HAADF) images (Fig. 2e, f) further indicate that the MoS$_x$-Au nanoparticles were effectively deposited on the TiO$_2$ surface. The corresponding energy-dispersive X-ray spectroscopy (EDS) mapping images (Fig. 2g–l) show that four Au nanoparticles exhibit uniform and same distribution with the Mo and S elements on the TiO$_2$ particles, unequivocally indicating that all the Au nanoparticles are selectively deposited on the MoS$_x$ surface via the self-assembly of S-Au. Ultraviolet-visible absorption spectra (UV-Vis) demonstrate a typical surface plasmon resonance (SPR) absorption of Au (ca. 540 nm) (Supplementary Fig. 5)[40,43,44], suggesting that Au was effectively deposited on the surfaces of both TiO$_2$ and TiO$_2$/MoS$_x$. Noticeably, the SPR signal in the TiO$_2$/MoS$_x$-Au appears relatively weaker than that of the TiO$_2$/Au, which can be attributed to the strong interaction between Au and S atoms. According to the inductively coupled plasma optical emission spectrometry (ICP-OES) results (Supplementary Table S1), the contents of Mo, S, and Au elements in the TiO$_2$/MoS$_x$-Au photocatalyst are 0.62, 0.8, and 3.34 wt%, respectively, indicating the presence of MoS$_x$ and Au in the TiO$_2$/MoS$_x$-Au system. These results

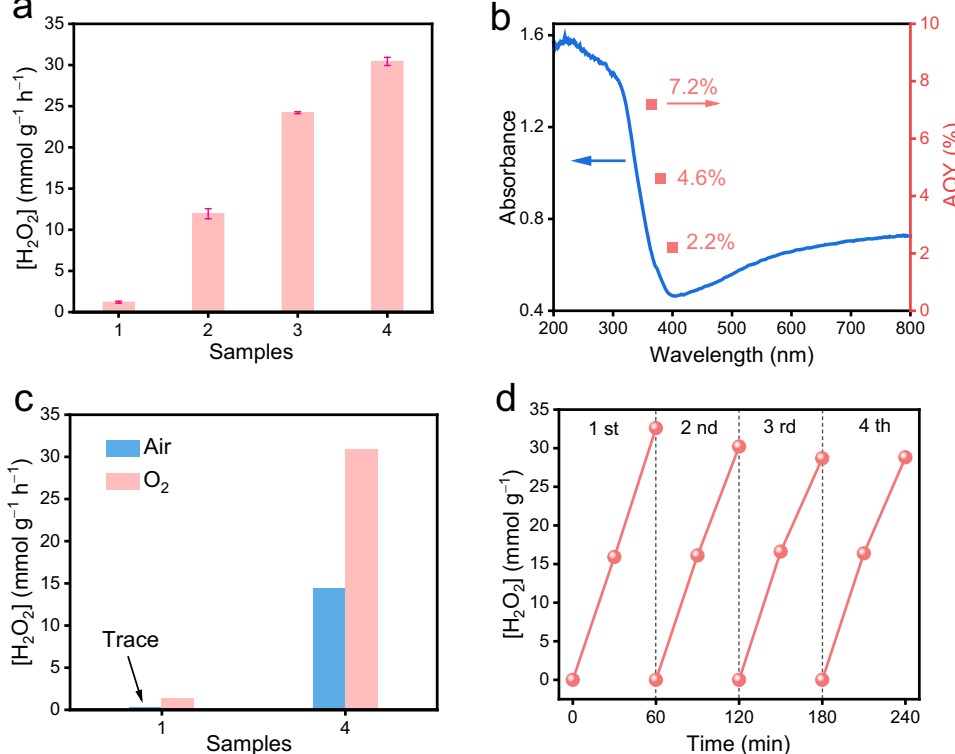

**Fig. 3 | Photocatalytic H₂O₂-production activities and stability. a** Photocatalytic H₂O₂-production performance for different samples in an ethanol-water solution (10% vol.): (1) TiO₂, (2) TiO₂/MoS$_x$, (3) TiO₂/Au, and (4) TiO₂/MoS$_x$-Au. The error bars (mean ± standard deviation) were calculated based on three independent photocatalytic experiments. **b** The AQY of H₂O₂ production as a function of wavelength on TiO₂/MoS$_x$-Au (red dot), and the UV-vis absorbance spectrum (blue curve). **c** The photocatalytic H₂O₂-production performance over (1) TiO₂ and (4) TiO₂/MoS$_x$-Au in the O₂-saturated and air condition. **d** Recycling H₂O₂-production performance of TiO₂/MoS$_x$-Au.

(HRTEM, UV-Vis, and ICP-OES) collectively support the selective deposition of Au on the MoS$_x$ surface.

## Photocatalytic performance

The photocatalytic activities for H₂O₂ production were conducted in an O₂-saturated ethanol solution under Xe lamp irradiation. As shown in Fig. 3a, TiO₂ exhibits a low H₂O₂-production rate of 1.20 mmol g⁻¹ h⁻¹. However, the introduction of Au nanoparticles onto the TiO₂ surface leads to an improved H₂O₂-production rate (24.22 mmol g⁻¹ h⁻¹) for the resulting TiO₂/Au. With the further incorporation of MoS$_x$ cocatalyst into TiO₂/Au, the TiO₂/MoS$_x$-Au shows a significant enhancement in the photocatalytic H₂O₂-production activity. Further investigation indicated that the H₂O₂-production activity of TiO₂/MoS$_x$-Au photocatalysts is dependent on the Au amount (Supplementary Fig. 6). When the Au is precisely maintained at 3%, the resulting TiO₂/MoS$_x$-Au sample exhibits the highest H₂O₂-production rate with a value of 30.44 mmol g⁻¹ h⁻¹ (Fig. 3a and Supplementary Fig. 6), which is 25.4 and 1.3 times higher than that of TiO₂ and TiO₂/Au, respectively. However, increasing the Au content beyond this point to 5% leads to a reduction in H₂O₂ yield due to the light-shielding effect. According to the wavelength-dependent H₂O₂-evolution activity (Fig. 3b), the AQY of TiO₂/MoS$_x$-Au achieves an impressive value of 7.2% at 365 nm. The present H₂O₂-evolution performance is much higher than most reported results in TiO₂-based photocatalysts or other inorganic photocatalysts (Supplementary Table S2). To investigate the application potential of present TiO₂/MoS$_x$-Au, its photocatalytic H₂O₂-evolution performance was further tested under different conditions. As exhibited in Fig. 3c and Supplementary Fig. 7, it is clear that there is almost no H₂O₂ generation over TiO₂ in the air environment. In contrast, the H₂O₂ yield in the TiO₂/MoS$_x$-Au still maintains a high

concentration (14.45 mmol g⁻¹ h⁻¹), indicating the great application potential of the TiO₂/MoS$_x$-Au photocatalyst. Besides, no significant decrease in the H₂O₂ concentration is observed after four cycles of photocatalytic reaction (Fig. 3d), revealing the robust reusability of the TiO₂/MoS$_x$-Au. From the above results, it can be concluded that the introduction of MoS$_x$ mediator into TiO₂/Au can effectively improve the photocatalytic H₂O₂-production activity.

## Photocatalytic mechanism of TiO₂/MoS$_x$-Au

Considering the boosted H₂O₂-production activity, the effect of the MoS$_x$ mediator on the Au electronic structure is primarily investigated by the first-principles calculations and XPS technology. For comparison, three slab models of Au, MoS$_x$, and MoS$_x$-Au are reasonably selected and optimized (Supplementary Fig. 8). Based on the optimized models, the work functions (Φ) of MoS$_x$ (001) and Au (111) are calculated to be 5.86 and 5.20 eV, respectively (Fig. 4a, b). In this case, when Au is loaded onto the MoS$_x$ surface, free electrons would migrate inevitably from Au nanoparticles to MoS$_x$ (Supplementary Fig. 9)[45,46], thus inducing the formation of electron-deficient Au$^{δ+}$ sites. This electron transfer is further substantiated by examining the local charge density difference and the corresponding planar-averaged electron density difference (Fig. 4c, d)[47]. Obviously, the MoS$_x$-Au cocatalyst shows a distinct electron-enriched region on the MoS$_x$ side, while positive charges predominantly accumulate on Au atoms, leading to the production of an electron-deficient Au$^{δ+}$ layer (Fig. 4d). To quantify the above charge transfer between Au and MoS$_x$, Bader charge calculation was performed and shown in Fig. 4e. Clearly, they indicate a more negative charge density for S and Mo atoms in MoS$_x$-Au compared to pure MoS$_x$. Conversely, the charge density of Au atoms is slightly increased (+0.07) to produce Au$^{δ+}$ active sites in the MoS$_x$-Au

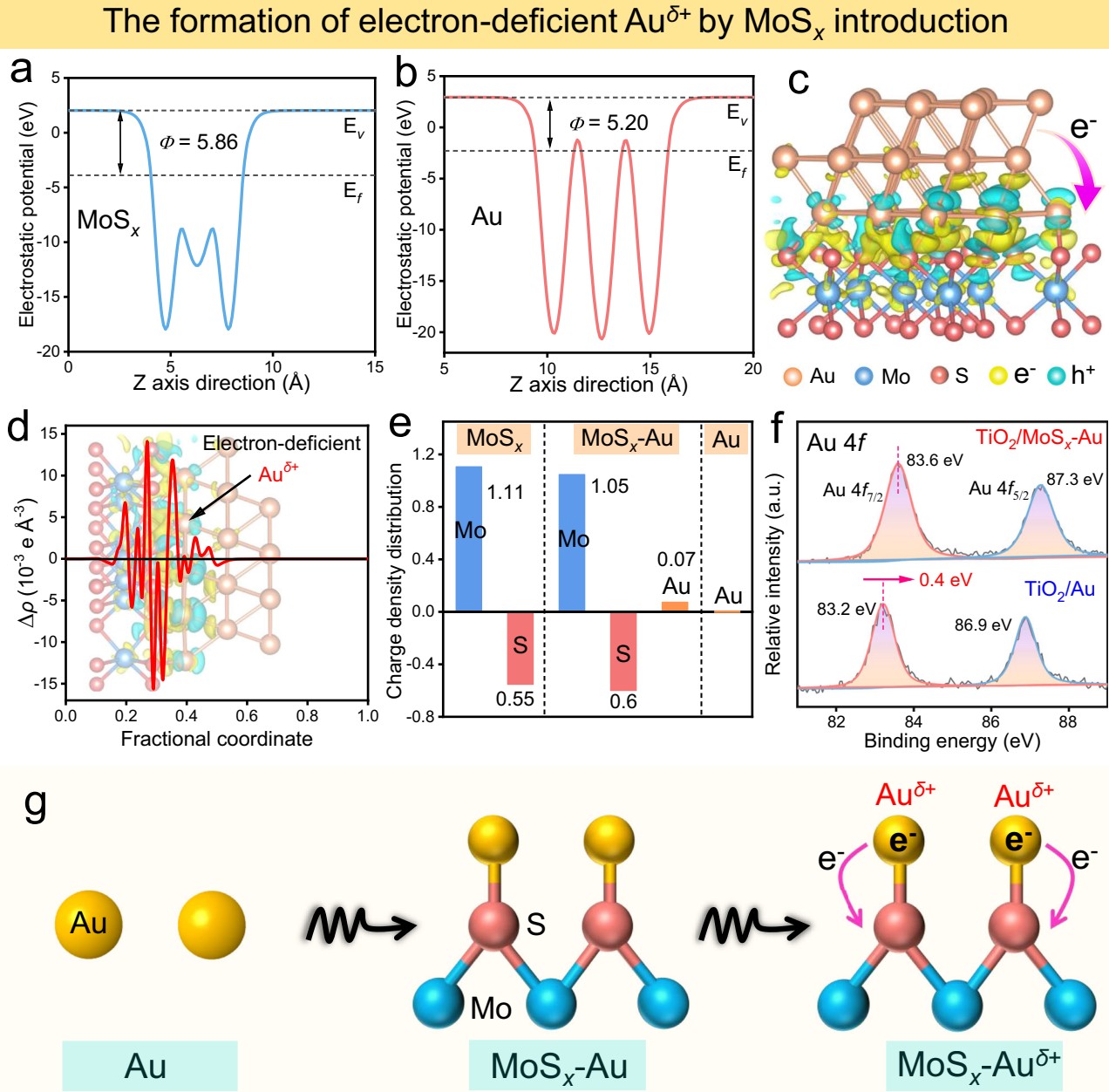

**Fig. 4 | MoS$_x$-induced electron-deficient Au$^{\delta+}$ formation and mechanism.**
**a, b** Calculated average potential profiles of MoS$_x$ and Au. **c** The local charge density difference of MoS$_x$-Au, where the light yellow and cyan areas represent electron accumulation and depletion, respectively. **d** Planar-averaged electron density difference $\Delta\rho(z)$ in MoS$_x$-Au. **e** The charge density distributions of MoS$_x$, MoS$_x$-Au, and Au. **f** High-resolution XPS spectra of Au 4$f$ in the TiO$_2$/Au and TiO$_2$/MoS$_x$-Au. **g** Schematic illustration of the formation of electron-deficient Au$^{\delta+}$ sites in the MoS$_x$-Au cocatalyst.

cocatalyst. The formation of above electron-deficient Au$^{\delta+}$ can further be verified by XPS analysis (Fig. 4f). Compared with the TiO$_2$/Au, a clear shift of binding energy (from 83.2 to 83.6 eV, $\Delta$ = 0.4 eV) to a higher value is observed for Au 4$f$ in the TiO$_2$/MoS$_x$-Au. In addition, the XPS peaks of S 2$p$ (162.0 eV) and Mo 3$d$ (231.6 eV) shift to lower values ($\Delta$= 0.6 eV for S 2$p$, $\Delta$= 0.3 eV for Mo 3$d$) than those of TiO$_2$/MoS$_x$ (Supplementary Fig. 10), indicating the efficient electron transfer from Au to MoS$_x$. The above electron transfer can also cause a slight change of binding energy for the Ti element (Supplementary Fig. 11). Unquestionably, both DFT calculation and experimental results strongly support that the introduction of MoS$_x$ mediator can effectively modulate the Au electronic structure to induce the formation of electron-deficient Au$^{\delta+}$ sites in the MoS$_x$-Au cocatalyst (Fig. 4g).

The generation of electron-deficient Au$^{\delta+}$ impacts the O$_2$ adsorption capability of TiO$_2$/MoS$_x$-Au photocatalyst, as evidenced by various analyses: O$_2$ adsorption energy, crystal orbital Hamilton population (COHP), bonding distance analysis, and partial density of states (PDOS) calculations. Based on the optimized models in Fig. 5a and Supplementary Fig. 12, the O$_2$ adsorption energies ($E_a$) of Au sites before and after MoS$_x$ introduction were first calculated (Fig. 5b). Clearly, the electron-deficient Au$^{\delta+}$ sites in the MoS$_x$-Au cocatalyst exhibit more negative adsorption energy ($E_a$ = −0.14 eV) than pure Au ($E_a$ = 0.48 eV), indicating stronger O$_2$ adsorption capability[48]. Further evidence of this enhanced adsorption is provided by COHP calculations[49], revealing a smaller integrated COHP value (−1.40) for Au-O$_{ads}$ bonds in MoS$_x$-Au than in pure Au (−1.38), and shorter bond length (2.01 Å) compared to pure Au (2.10 Å), signifying

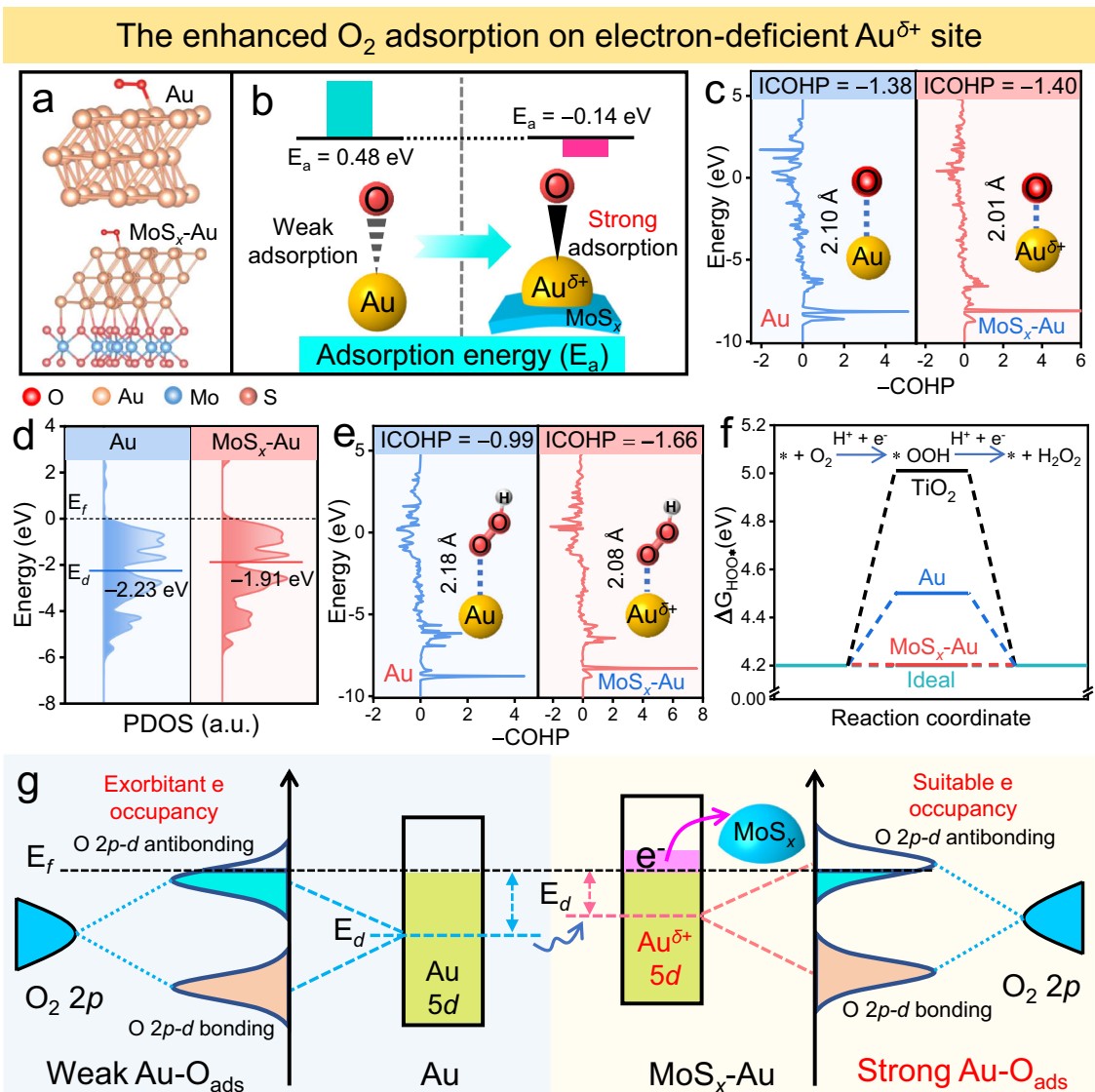

**Fig. 5 | Modifying the electron structure of Au cocatalyst to decrease the antibonding-orbital occupancy of Au-O$_{ads}$ for enhancing Au-O$_{ads}$ bonds.** **a** Optimized configurations for O$_2$ adsorption on Au and MoS$_x$-Au, and **b** the corresponding adsorption energies and schematic illustration about the improved O$_2$-adsorption ability. **c** COHP analyses of O$_2$ adsorption on Au and MoS$_x$-Au. **d** PDOS diagrams of Au 5$d$ orbitals in Au and MoS$_x$-Au. **e** COHP analyses of *OOH adsorption on Au and MoS$_x$-Au (* represents active site). **f** $\Delta G_{HOO*}$ over TiO$_2$, Au and MoS$_x$-Au. **g** Schematic diagram about the enhanced O$_2$ adsorption on Au sites by decreasing antibonding-orbital occupancy via raising its $d$-band center.

stronger Au-O$_{ads}$ bonding in electron-deficient Au$^{\delta+}$ sites (Fig. 5c)[50]. In this case, the $d$-band center mechanism helps explain this enhanced Au-O$_{ads}$ bonding in the MoS$_x$-Au[51]. As depicted in Fig. 5d, g, the $d$-band center of Au 5$d$ in pure Au cocatalyst is −2.23 eV, significantly far away from E$_f$. Consequently, the lower $d$-band center causes exorbitant antibonding-orbital occupancy, leading to a weak Au-O$_{ads}$ bond. However, when Au is loaded on the MoS$_x$ surface, the resulting $d$-band center of Au 5$d$ in the MoS$_x$-Au is closer to the E$_f$ (−1.91 eV) than that of pure Au suggesting that modulating the electron structure of the Au cocatalyst to form electron-deficient Au$^{\delta+}$ sites can significantly elevate the $d$-band center (Fig. 5g). In this case, when O$_2$ is adsorbed on the electron-deficient Au$^{\delta+}$ sites, the antibonding-orbital occupancy of Au-O$_{ads}$ is decreased, resulting in a reinforced O$_2$ adsorption. Therefore, the presence of MoS$_x$ cocatalyst in the MoS$_x$-Au can effectively raise the $d$-band center of Au$^{\delta+}$ sites for improved O$_2$ adsorption, which is one of the essential steps for the following H$_2$O$_2$ production.

It is worth emphasizing that a suitable Au-O$_{ads}$ bond can effectively contribute to the formation of Au-OOH$_{ads}$ intermediate, thus

greatly improving the selectivity and activity of photocatalytic H$_2$O$_2$ production (Supplementary Fig. 13). Hence, to explore the effect of improved Au-O$_{ads}$ bonds on the formation of Au-OOH$_{ads}$ intermediate in MoS$_x$-Au cocatalyst, COHP calculation was carried out. As illustrated in Fig. 5e, both COHP value (−1.66) and bond length (2.08 Å, inset) of Au-OOH$_{ads}$ in MoS$_x$-Au are smaller than those in pure Au (−0.99 and 2.18 Å), forcefully manifesting that the MoS$_x$-Au possess stronger Au-OOH$_{ads}$ bonds, which is beneficial to improving the selectivity of photocatalytic H$_2$O$_2$ production. The above selectivity and activity of photocatalytic H$_2$O$_2$ production on MoS$_x$-Au cocatalyst can further be confirmed by the free energy of *OOH intermediate ($\Delta G_{HOO*}$) based on the optimized models in Supplementary Fig. 14[22]. As shown in Fig. 5f, the $\Delta G_{HOO*}$ values on TiO$_2$ and Au sites are estimated to be 5.01 and 4.5 eV, respectively, which are significantly higher than the ideal $\Delta G_{HOO*}$ value (4.2 eV)[52]. These results suggest that TiO$_2$ and Au have relatively weak adsorption and easier detachment for *OOH intermediates, leading to sluggish interfacial H$_2$O$_2$-production kinetics (Supplementary Fig. 13a). In contrast, the electron-deficient Au$^{\delta+}$ sites

**Fig. 6 | Photogenerated electron transfer mechanism and dynamics. a**, **b** KPFM images and the corresponding surface potential profiles of the TiO$_2$/MoS$_x$-Au in the dark and light illumination, a 365 nm-LED light as the light source. **c** ISI-XPS spectra of Au 4$f$ for TiO$_2$/MoS$_x$-Au before and after light illumination. Pseudocolor plots of **d** TiO$_2$ and **e** TiO$_2$/MoS$_x$-Au (GSB represents ground-state bleaching). **f**, **g** Femtosecond transient absorption spectra of TiO$_2$ and TiO$_2$/MoS$_x$-Au within 20 ps. All data were obtained under excitation of 330 nm and optical power of 600 µW cm$^{-2}$. Schematic illustration of the decay pathways of photogenerated electrons in **h** TiO$_2$ and **i** TiO$_2$/MoS$_x$-Au.

in the MoS$_x$-Au cocatalyst showcase the best $\Delta G_{HOO*}$ values (4.2 eV), aligning with the ideal energy for *OOH adsorption and facilitating rapid H$_2$O$_2$ generation (Supplementary Fig. 13b).

In addition to enhancing O$_2$ adsorption for efficient H$_2$O$_2$ production, the MoS$_x$-Au cocatalyst also promotes the rapid transfer of photogenerated electrons in the TiO$_2$/MoS$_x$-Au photocatalyst, which is supported by the subsequent Kelvin probe force microscopy (KPFM) and in situ XPS[53,54]. A scanning probe microscopy (SPM) system with KPFM was employed to analyze the distribution and transfer pathways of photogenerated electron-hole pairs in photocatalysts. The resulting AFM topography image, KPFM image, and the corresponding contact potential difference (CPD) profiles of TiO$_2$ and TiO$_2$/MoS$_x$-Au are shown in Fig. 6a, b, and Supplementary

Figs. 15 and 16. Obviously, the TiO$_2$/MoS$_x$-Au particles are observed, and a line scan across the above sample pre- and post-light irradiation is used to evaluate the CPD change. Upon light irradiation, the CPD value of TiO$_2$ shows a slight increase from −14.7 to 8.6 mV owing to the spontaneous transfer of photogenerated holes onto the TiO$_2$ surface (Supplementary Fig. 16)[55]. After the loading of MoS$_x$-Au, the TiO$_2$/MoS$_x$-Au exhibits an obvious CPD value increase of about 185.4 mV (from −51.3 to 134.1 mV) during light irradiation, accompanied by a color change from blue to red owing to enhanced hole accumulation on the TiO$_2$ surface (Fig. 6a, b), strongly indicating that the photogenerated electrons are efficiently transferred from TiO$_2$ to MoS$_x$-Au cocatalyst[56]. To further validate the above transfer of photogenerated electrons and their enrichment on the Au active sites of

TiO$_2$/MoS$_x$-Au, in situ XPS was performed (Fig. 6c). The peaks of Au 4$f_{7/2}$ and Au 4$f_{5/2}$ in the TiO$_2$/MoS$_x$-Au are remarkably shift toward lower binding energies (from 83.6 eV to 83.5 eV, $\Delta$ = 0.1 eV) upon light irradiation, suggesting that the photogenerated electrons are directionally transferred from TiO$_2$ to MoS$_x$-Au and mainly enriched on the electron-deficient Au$^{\delta+}$ sites, thereby promoting the photocatalytic H$_2$O$_2$-production kinetics[57].

For a comprehensive understanding of electron-transfer dynamics in the TiO$_2$/MoS$_x$-Au, femtosecond transient absorption spectroscopy (fs-TAS) was carefully performed[58]. As shown in Fig. 6d, e and Supplementary Fig. 17a, a typical photobleaching peak (~380 nm) is displayed in the pseudocolor plots of TiO$_2$, TiO$_2$/Au, and TiO$_2$/MoS$_x$-Au. These signals are assigned to the ground-state bleaching (GSB), which reflects the excited state relaxation[59]. Further monitoring of the GSB signal (380 nm) within 20 ps reveals stronger intensity in the TiO$_2$/Au (Supplementary Fig. 17b) and TiO$_2$/MoS$_x$-Au (Fig. 6g) compared to TiO$_2$ (Fig. 6f), indicating enhanced electron enrichment in both TiO$_2$/Au and TiO$_2$/MoS$_x$-Au[39]. Further analysis of the interfacial electron transfer involved fitting decay kinetics within 25 ps at 380 nm using biexponential equations, with fitting results of normalized curves shown in Supplementary Fig. 18 and Table S3. The short-lived $\tau_1$ corresponds to the electron trapping by electron trapping state (e-TS), while the long-lived $\tau_2$ is related to the interfacial electron transfer from TiO$_2$ to cocatalyst. Meanwhile, $A_1$ and $A_2$ represent the decay proportion of photogenerated electrons during the electron trapping and transfer, respectively. Obviously, the TiO$_2$ primarily undergoes a short-lived process within 25 ps under irradiation, and the corresponding $\tau_1$ is 1.63 ps, which is primarily attributed to electron trapping in the e-TS (Fig. 6h). Interestingly, the $\tau_1$ value in the TiO$_2$/Au and TiO$_2$/MoS$_x$-Au significantly decreases to 0.36 and 0.92 ps, respectively, suggesting rapid transfer of a portion of photogenerated electrons from TiO$_2$ to Au ($\tau_2$ = 5.88 ps) and MoS$_x$-Au ($\tau_2$ = 7.10 ps) cocatalysts (Fig. 6i). Noticeably, the TiO$_2$/MoS$_x$-Au exhibits a larger $A_2$ value ($A_2$ = 39.6%) compared to the TiO$_2$/Au ($A_2$ = 32.3%), indicating more effective transfer of photogenerated electrons from TiO$_2$ to Au facilitated by the MoS$_x$ mediator[60]. The improved electron transfer on TiO$_2$/MoS$_x$-Au is well consistent with the results of photoelectrochemical and transient-state photoluminescence (TRPL) (Supplementary Fig. 19). These above results provide concertedly evidence that the MoS$_x$-Au cocatalyst serves as an efficient platform for rapid transfer of photogenerated electrons to engage in the subsequent H$_2$O$_2$-production reaction at the electron-deficient Au$^{\delta+}$ sites (Fig. 6i), thus achieving high photocatalytic H$_2$O$_2$ yields.

Overall, a strategy of electronic structure modification for the Au cocatalyst has been proposed to effectively reinforce the Au-O$_{ads}$ bonding at electron-deficient Au$^{\delta+}$ sites within the MoS$_x$-Au cocatalyst, which can achieve an enhanced O$_2$ adsorption for fast H$_2$O$_2$-production kinetics. As a result, an exceptional H$_2$O$_2$-production rate of 30.44 mmol g$^{-1}$ h$^{-1}$ has been achieved in the resulting TiO$_2$/MoS$_x$-Au, which is 25.4 and 1.3 times higher than that of TiO$_2$ and TiO$_2$/Au, respectively. Theoretical simulations and experimental results consistently support the notion that the introduction of MoS$_x$ mediator induces the formation of electron-deficient Au$^{\delta+}$ active sites in the MoS$_x$-Au cocatalyst by free-electron transfer from the Au cocatalyst to MoS$_x$, which decreases the antibonding-orbital occupancy of the Au-O$_{ads}$, thereby enhancing the O$_2$-adsorption ability to realize efficient H$_2$O$_2$-production performance. In addition, the MoS$_x$-Au cocatalyst can also provide an efficient platform for the rapid transfer and enrichment of photogenerated electrons from TiO$_2$, leading to a distinct improvement of photocatalytic H$_2$O$_2$-production activity for the TiO$_2$/MoS$_x$-Au. This work emphasizes a feasible strategy for optimizing the O$_2$-adsorption strength to efficiently accelerate H$_2$O$_2$-production kinetics, offering a very promising approach for the rational design of electronic structure for efficient artificial photosynthesis.

## Methods

### Preparation of TiO$_2$/MoS$_x$ photocatalyst

TiO$_2$ photocatalyst (P25) was calcined at 550 °C for 2 h in a muffle furnace before being used. The TiO$_2$/MoS$_x$ sample was synthesized by one-step lactic acid-induced method, as schematically demonstrated in Supplementary Fig. 1. First, 624 μL (NH$_4$)$_2$MoS$_4$ (0.02 mol/L) solution was dropped into 160 mL lactic acid solution (10 vol%) under stirring. In this case, the H$^+$ was released from lactic acid and would induce the transformation of MoS$_4^{2-}$ into MoS$_x$ colloidal nanoparticles. Subsequently, the as-prepared TiO$_2$ nanoparticles (0.1 g) were dispersed into the above solution. After stirring for 2 h, a brown product was collected by centrifugation and washing with deionized water and ethanol. Finally, the obtained product was dried at 80 °C for 12 h. The resulting brown powder was denoted as TiO$_2$/MoS$_x$. In addition, the pure MoS$_x$ product was also obtained by a similar synthesis route of the above TiO$_2$/MoS$_x$ in the absence of TiO$_2$.

### Preparation of TiO$_2$/MoS$_x$-Au photocatalyst

The MoS$_x$-Au modified TiO$_2$ photocatalyst (TiO$_2$/MoS$_x$-Au) was synthesized by a two-step route, including the initial deposition of MoS$_x$ on the TiO$_2$ surface and the subsequently selective photodeposition of Au onto the MoS$_x$ surface. First, 0.1 g of the MoS$_x$/TiO$_2$ was mixed with 80 mL ethanol aqueous solution (20 vol%) in a 100 mL three-necked flask. Then, a known amount of chloroauric acid (0.1 mol/L, HAuCl$_4$·4H$_2$O) was added. After the above system was evacuated with N$_2$ for 20 min and then irradiated with a 300 W Xenon lamp for 1 h, the resultant suspension was collected by centrifugated, rinsed, and dried at 80 °C for 12 h to obtain the final TiO$_2$/MoS$_x$-Au. To investigate the effect of Au amount on the structure and photocatalytic performance, the amount of Au in the TiO$_2$/MoS$_x$ was controlled to be 1, 1.5, 2, 3, and 5 wt%, respectively, and the resultant sample was referred to be TiO$_2$/MoS$_x$-Au-$X$% ($X$ is the amount of Au).

### Preparation of TiO$_2$/Au photocatalyst

Au nanoparticle-loaded TiO$_2$ (TiO$_2$/Au) was prepared by a photodeposition method. First, 0.1 g of the TiO$_2$ was dispersed into a 100 mL of ethanol aqueous solution (20 vol%). Subsequently, a known amount of chloroauric acid (0.1 mol/L, HAuCl$_4$·4H$_2$O) was added dropwise to the above mixture solution. Before irradiation, the above system was bubbled with N$_2$ for 20 min and then irradiated for 1 h. Finally, a light-purple product was collected by centrifugated, rinsed, and dried at 60 °C for 12 h. The resulting powder was denoted as TiO$_2$/Au.

### Characterization

The microstructure of the samples was characterized by transmission electron microscopy (TEM; Thermal Fisher Talos F200X). X-ray photoelectron spectroscopy (XPS) was performed on a Thermo Scientific ESCALA 210 XPS spectrometer system (USA) with 300 W Al Kα radiation to survey the elemental composition and valence states of these photocatalysts. The X-ray diffraction (XRD) patterns of the samples were obtained on an X-ray diffractometer (Shimadzu XRD-6100) with Cu Kα radiation. The elemental content was performed by inductively coupled plasma optical emission spectrometry (ICP-OES). The ultraviolet-visible spectra (UV-vis DRS) were obtained on a UV-vis spectrophotometer (UV-2600i, Shimadzu, Japan). Time-resolved photoluminescence (TRPL) spectra were acquired on a fluorescence lifetime spectrophotometer (FLS 1000, Edinburgh, UK). The photo-irradiated Kelvin probe force microscopy (KPFM) (SPM-9700, Shimadzu, Japan) was carried out to test the contact potential difference of the samples.

### Photocatalytic H$_2$O$_2$ production test

Photocatalytic H$_2$O$_2$-production activity was examined in an O$_2$-saturated aqueous solution with ethanol as a hole scavenger, and a 300 W Xenon arc lamp was selected as the light source. First, 10 mg of

as-prepared photocatalyst and 100 mL of ethanol solution (10 vol%) were mixed in a 100 mL three-necked flask reactor. Before irradiation, the system was purged with oxygen for 30 min to obtain an $O_2$-saturated solution. During the photocatalytic $H_2O_2$-production test, 1 mL of solution was sampled from the reactor. Finally, the $H_2O_2$ concentration was examined via an iodometry method by using a UV-visible spectrophotometer (UV-1240, Japan). The concentration of $H_2O_2$ was calculated by the equation ($y = 0.00771x + 0.0218$), and the reaction mechanism was shown in Eq. (1). The absorbance of $I_3^-$ at 350 nm can be recorded by UV-vis spectroscopy.

$$H2O_2 + 3I^- + 2H^+ \rightarrow I_3^- + 2H_2O \tag{1}$$

### Photoelectrochemical measurements
The photoelectrochemical (PEC) properties were assessed using an electrochemical workstation (CHI 760E, China) within a three-electrode system. The working electrode was prepared by applying the photocatalyst onto a 1.0 cm$^2$ FTO glass substrate. The Ag/AgCl (in a saturated KCl solution) and Pt foil were designated as the reference and counter electrodes, respectively. The PEC assessments were carried out in a 0.5 M $Na_2SO_4$ solution using a 300 W Xenon arc lamp for illumination.

### Average decay time ($\tau_{average}$) calculation
The decay curves of as-prepared samples from the TRPL can be effectively fitted using the following biexponential Eq. (2), and the fluorescent lifetime ($\tau_a$) is calculated by Eq. (3).

$$A_{(t)} = A_{(0)} + A_1 \exp(-t/\tau_1) + A_2 \exp(-t/\tau_2) \tag{2}$$

$$\tau_a = (A_1\tau_1{}^2 + A_2\tau_2{}^2)/(A_1\tau_1 + A_2\tau_2) \tag{3}$$

where $A_1$ and $A_2$ represent the weight factors, while $\tau_1$ and $\tau_2$ are the short and long fluorescent lifetimes, respectively.

### Apparent quantum yield (AQY) calculation
The AQY measurement was conducted in an $O_2$-saturated aqueous solution with ethanol as a hole scavenger by utilizing a 300 W Xenon arc lamp as the light source. In detail, the as-prepared photocatalyst (10 mg) and ethanol solution (100 mL, 10 vol%) were mixed in a 100 mL three-necked flask reactor, which was oxygenated for 30 min to obtain the $O_2$-saturated solution.

The apparent quantum yields for $H_2O_2$ were calculated from the following Eq. (4):

$$\eta = \frac{N_e}{N_p} \times 100\% = \frac{2 \times M \times N_A \times h \times c}{S \times P \times t \times \lambda} \times 100\% \tag{4}$$

where $M$ represents the amount of produced $H_2O_2$ molecules (mol), $N_A$ is the Avogadro constant ($6.022 \times 10^{23}$/mol), $h$ is the Planck constant ($6.626 \times 10^{-34}$ J s), $c$ is the speed of light ($3 \times 10^8$ m/s), $S$ is the irradiation area (15.83 cm$^2$), $P$ is the average intensity of irradiation (365 nm, 13.35 mW/cm$^2$), $t$ is the irradiation time (s), and $\lambda$ is the wavelength of the incident monochromatic light (nm).

### Ultrafast transient absorption (TA) tests
Femtosecond transient absorption spectra of the as-prepared photocatalysts were obtained on a pump-probe system (Helios, Ultrafast System) with a maximum time delay of ~8 ns using a motorized optical delay line under ambient conditions. The 330 nm-pump pulses (600 µW average at tested samples) were generated by the 1 kHz regenerative amplifier (Coherent Libra, 800 nm, 35 fs, 5 mJ) in an optical parametric amplifier (OPerA Solo), seeded with a mode-locked Ti: sapphire oscillator (Coherent Vitara, 800 nm, 80 MHz) and pumped with an LBO laser (Coherent Evolution-50C, 1 kHz system). To generate the white light from 320 to 750 nm, the 800 nm-femtosecond pluses were pumped by a constantly rotating sapphire crystal.

### Computational details
The density functional theory (DFT) calculations were performed using the Vienna Ab initio Simulation Package (VASP) with the generalized gradient approximation (GGA) employing the revised Perdew–Burke–Ernzerhof (PBE) functional for the exchange-correlation interaction. The convergence threshold for total energy converged within $10^{-4}$ eV/atom, and the average force was 0.01 eV/Å. Grid integration utilized a cutoff energy of 450 eV, and projector-augmented wave (PAW) potentials characterized the ion cores. To rationalize the calculation, unsaturated sulfur (S) atoms were obtained on the edge of $MoS_x$ model by creating a vacuum layer in the $y$-axis direction, which displayed a similar local coordination structure as amorphous $MoS_x$ cocatalyst. Moreover, ($3 \times 3 \times 1$) Monkhorst-Pack grids and ($2 \times 2$) surface cells were used for oxygen adsorption. The adsorption energy $E_{ads}$ was defined as $E_{ads} = E_{total} - E_{surface} - E_{O2}$, where $E_{total}$, $E_{surface}$, and $E_{O2}$ represent the energy of adsorption configurations, the energy of metallic surfaces, and the energy of molecular $O_2$, respectively. In addition, the ΔG of HOO∗ intermediate on the surface was calculated by the equation $G = E + ZPE - TS$, where $E$ is the total energy, $ZPE$ is the zero-point energy, $T$ is the temperature (298.15 K), and $S$ is the entropy. Several configurations of the adsorbed models were considered in the simulation, and the most favorable ones are presented based on the adsorption energy.

## Data availability
Data is available from the authors on request. All data generated in this study are provided in the Source data file. Source data are provided with this paper.

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

## Acknowledgements

This work was supported by the National Natural Science Foundation of China (22278324 (J.Y.), 52073223 (J.Y.), U22A20147 (H.Y.) and 22261142666 (J.Y.)) and the Natural Science Foundation of Hubei Province of China (2022CFA001 (H.Y.)). We thank the Faculty of Materials Science and Chemistry, China University of Geosciences (CUG), Wuhan for its TEM facilities and the data analysis of Dr. Mingxing Gong.

## Author contributions

X.Z., D.G., and H.Y. conceived and designed the experiments. X.Z., D.G., and B.Z. carried out the synthesis, characterizations, and theory calculations of the materials. X.Z. and D.G. carried out the photocatalytic test. X.Z., B.C., J.Y., and H.Y. contributed to data analysis. J.Y. and H.Y. supervised the project. X.Z. and H.Y. wrote the manuscript. B.C., J.Y., and H.Y. revised and reviewed the manuscript. All authors discussed the results and commented on the manuscript.

## Competing interests

The authors declare no competing interests.
