## [Peer Review File · Nature Communications]

REVIEWER COMMENTS

Reviewer #1 (Remarks to the Author):

This manuscript reports an electron discharging strategy of Au active sites to strengthen the oxygen adsorption energy and boost photocatalytic H₂O₂ generation. MoS_x was used as an electron mediator, which resulted in the formation of defective Au^{δ+} active sites. The TiO₂/MoS_x-Au photocatalyst achieves a significantly boosted H₂O₂-production rate of 30.44 mmol g⁻¹ h⁻¹, which is 1.3 times higher than that of TiO₂/Au. With the increasing attention and importance of H₂O₂ synthesis reactions and the high H₂O₂ production rate of the photocatalyst, the manuscript possesses sufficient impact to be published in nature communications. However, some points must be proven with supplementary data to back up the key arguments of the manuscript. Therefore, the reviewer thinks this manuscript should be published with some major revisions.

Comments 1.: Are all of the Au nanoparticles selectively deposited on MoS_x? Please provide a more solid evidence. X-ray spectroscopies such as EXAFS might help. Also, please provide further details and reinforce references regarding decrease of not only SPR signals but overall absorbance signals of TiO₂/MoS_x-Au in the UV-DRS data.

Comments 2.: Please provide the production rate of H₂O₂ of all samples as a plot of time, for longer hours. As produced H₂O₂ accumulates over time, Au will participate in the catalytic degradation of H₂O₂, and the formation of Au^{δ+} sites might play a negative role, in this context. Also, in Fig.3(c), please provide the full data for all samples (1~4).

Comments 3.: ICP results indicate that relatively small amount of MoS_x is present compared to Au, but in Fig.4(e, f), a small charge density change in Mo and S atoms induce a relatively huge charge density change in Au. What is the reason?

Comments 4.: In accordance with comment3, for the total charge to be preserved, there might be some electronic structural changes in TiO₂. Please suggest further details about the electronic states of TiO₂, including Ti 2p XPS spectra. Presence of TiO₂ is generally overlooked throughout the manuscript.

Comments 5.: Does this electron discharging strategy work for other non-plasmonic metals such as Pd? This could not only prove the versatility of the suggested charge re-distribution strategy, but also separate the possible effect of surface plasmon resonance.

Comments 6.: In Table S2, please also add some references that use Au metals as cocatalysts (preferably more than 5). Some are also mentioned in the introduction section.

Reviewer #2 (Remarks to the Author):

This work by Xidong Zhang et al. reports on photocatalytic production of H₂O₂ with a rate that is very high indeed compared to the recent literature. The authors attribute this success partial charge transfer from Au nanorattles (at which oxygen reaction is supposed to take place) to underlying MoS_x due to the difference of their workfunctions. The preparation of the catalyst, its characterization, experimental conditions, and computational modelling details are described in sufficient detail. The work functions from the DFT calculations are in good agreement with published experimental values.

Concerning the results obtained at different catalysts, there is some confusion if one compares Figure 3a and the statements at different places of the text, in which the less efficient catalyst compositions are presented in different order:

87-88

The synthesized TiO₂/MoS_x-Au photocatalyst achieves a boosted H₂O₂-production rate of 30.44 mmol g⁻¹ h⁻¹, which is 25.4 and 1.3 times higher than that of 88 TiO₂ and TiO₂/Au, respectively.

167-169

the optimal photocatalytic H₂O₂ yield in TiO₂/MoS_x-Au reached to 30.44 mmol g⁻¹ h⁻¹ (Fig. 3a and Supplementary Fig. 6), which is 25.4 and 1.3 times higher than that of TiO₂/Au and TiO₂/MoS_x, respectively.

Supposing Fig. 3a is correct, the effect of MoS_x underlayer on the increase of Au activity is relatively weak (30%), while the authors see in it the main message of their work. On the other hand, Au/TiO₂ is already very active, compared to the literature. An important result presented in Supplementary Figure 6 but not discussed is virtual independence of rate on Au content changing 5-fold (from 1 to 5%). How is it possible that O₂ is supposed to be chemisorbed on Au but increased amount (surface?) of Au does not produce more H₂O₂. At the same time, the authors are sure that Au is at the surface of MoS_x:

the contents of Mo, S, and Au elements in the TiO₂/MoS_x-Au photocatalyst are 0.62, 0.8, and 3.34 wt%, respectively. Doubtlessly, these above results collectively support the selective deposition of Au on the MoS_x surface in the resulting TiO₂/MoS_x-Au photocatalyst.

This needs to be discussed and clarified.

In all, there is vast literature on H₂O₂ photocatalytic production at TiO₂/Au; which merits a better discussion than provided in the Introduction. The roles of Au and MoS_x underlayer remain not quite clear. I invite the authors to reconsider the interpretation of their results and reconsider the work for publication afterwards.

Reviewer #3 (Remarks to the Author):

The manuscript describes the synthesis of a TiO₂/MoS_x-Au composite and its use for photocatalytic H₂O₂ generation with ethanol as an electron donor. The authors claimed enhanced H₂O₂ generation by the creation of the three component systems. However, the manuscript contains several problems at the present form. Followings should be considered.

1) TiO₂/MoS_x is a well-known photocatalyst for enhancing charge separation (<https://doi.org/10.1016/j.apcatb.2021.120878>). TiO₂/Au is a well-known photocatalyst for H₂O₂ generation (<https://doi.org/10.1021/ja102651g>). Moreover, MoS_x/Au has also already been reported to behaves as a good photocatalyst for H₂O₂ generation (<https://doi.org/10.1016/j.jcat.2019.06.015>). Based on this situation, the present photocatalyst is just the combination of these reported materials. I therefore must say that the novelty of this manuscript is insufficient.

2) The biggest problem is Figure 3a. The activity of TiO₂/MoS_x-Au (4) is just a sum of TiO₂/MoS_x (2) and TiO₂/Au (3). This means that “effect of three component system is negligible.” The authors should reconsider this situation, which is the part that decisively affects the importance of this paper.

3) The basic problem of this manuscript is that the system requires ethanol as an electron donor. Several photocatalytic systems using sacrificial electron donors have been reported; many of them shows much higher activity than the present system, although the authors did not cite these papers at al.

Based on the above, I do not think the novelty and quality of this manuscript justify publication in this journal.

Responses to the reviewers' comments

Reviewer #1

This manuscript reports an electron discharging strategy of Au active sites to strengthen the oxygen adsorption energy and boost photocatalytic H₂O₂ generation. MoS_x was used as an electron mediator, which resulted in the formation of defective Au^{δ+} active sites. The TiO₂/MoS_x-Au photocatalyst achieves a significantly boosted H₂O₂-production rate of 30.44 mmol g⁻¹ h⁻¹, which is 1.3 times higher than that of TiO₂/Au. With the increasing attention and importance of H₂O₂ synthesis reactions and the high H₂O₂ production rate of the photocatalyst, the manuscript possesses sufficient impact to be published in nature communications. However, some points must be proven with supplementary data to back up the key arguments of the manuscript. Therefore, the reviewer thinks this manuscript should be published with some major revisions.

Comments 1: Are all of the Au nanoparticles selectively deposited on MoS_x? Please provide a more solid evidence. X-ray spectroscopies such as EXAFS might help. Also, please provide further details and reinforce references regarding decrease of not only SPR signals but overall absorbance signals of TiO₂/MoS_x-Au in the UV-DRS data.

Author Reply 1: We are very grateful for the reviewer's comment. In this work, we found that all the Au nanoparticles were selectively deposited on the MoS_x surface, which can be well demonstrated by the following facts. First, according to the synthetic mechanism, the Au nanoparticles were selectively deposited on the MoS_x surface of TiO₂/MoS_x photocatalysts by a photoinduced deposition route (Supplementary Fig. 1).

It is well known that the AuCl_4^- ions can be selectively adsorbed on the S active sites of MoS_x surface due to the strong interaction between S and Au elements when HAuCl_4 aqueous solution is added into the $\text{TiO}_2/\text{MoS}_x$ suspension. During the following illumination, the photogenerated electrons of TiO_2 can directionally transfer to the MoS_x cocatalyst, where the adsorbed AuCl_4^- ions can be effectively reduced to produce metallic Au nanoparticles on the MoS_x surface. Secondly, the analysis results in Fig. 2 in our manuscript strongly confirms the selective deposition of Au on the MoS_x surface to synthesize the $\text{TiO}_2/\text{MoS}_x$ -Au photocatalyst, in good agreement with the above synthetic strategy. Finally, to further determine whether all Au nanoparticles are selectively deposited on the MoS_x surface, we perform the further HRTEM analysis for different regions of $\text{TiO}_2/\text{MoS}_x$ -Au sample, which can provide direct and visual evidence of the selective deposition for Au nanoparticles. A typical HRTEM result of the $\text{TiO}_2/\text{MoS}_x$ -Au sample is shown in Fig. R1. It is found that there is some amorphous structure between the Au and TiO_2 nanoparticles in Fig. R1c, which can be attributed to the amorphous MoS_x , clearly suggesting the excellent coupling of Au on MoS_x surface. In Fig. R1e, in addition to two larger Au nanoparticles (ca. 5 nm), two smaller Au can also be observed on a TiO_2 particle surface. Their corresponding EDS mapping images show that the four Au nanoparticles exhibit uniform and same distribution with the Mo and S elements on the TiO_2 particles, unequivocally indicating that all the Au nanoparticles are selectively deposited on the MoS_x surface, which is independent of Au particle size.

X-ray absorption near edge structure (EXAFS) is usually applied to demonstrate the

single atom of noble metals. In this study, in addition to the Au-S bond, the Au-Au bond is also existed in the present TiO₂/MoS_x-Au photocatalyst. Herein, we used HRTEM image and its EDS mapping to confirm the above selective deposition of Au nanoparticles on the MoS_x surface.

In the UV-vis absorption spectra in Supplementary Fig. 5, it is found that the visible-light absorption of TiO₂/MoS_x-Au is slightly lower than that of the TiO₂/Au, which can be attributed to the different chemical environment for Au nanoparticles. Obviously, the TiO₂/Au photocatalyst shows overall visible-light absorption and a typical SPR signals about 500-600 nm owing to the deposition of Au nanoparticles on TiO₂ surface. After the Au nanoparticles were selectively deposited on the MoS_x surface, the resulting TiO₂/MoS_x-Au sample shows a slightly decreased visible-light absorption. To further confirm the above results, we re-prepare the TiO₂, TiO₂/MoS_x, TiO₂/Au and TiO₂/MoS_x-Au samples, and their corresponding UV-vis results are shown in Fig. R2. Clearly, a slight decrease of the visible-light absorption for the TiO₂/MoS_x-Au can also be observed compared with the TiO₂/Au. Considering the important effect of chemical environment on SPR signals of Au nanoparticles, we believed that the slightly decreased absorption in the visible-light absorption is attributed to the formation of strong S-Au interaction in the TiO₂/MoS_x-Au, in good agreement with the previous work¹.

Fig. R1 (a, b) TEM, (c) HRTEM, (d, e) HAADF-STEM, and EDS mapping images of $\text{TiO}_2/\text{MoS}_x\text{-Au}$.

Fig. R2 UV-vis absorption spectra of various samples.

References:

1. Yadav, V. et al. Modulating the structure and hydrogen evolution reactivity of metal

chalcogenide complexes through ligand exchange onto colloidal Au nanoparticles. *ACS Catal.* **10**, 13305-13313 (2020).

Our modification to the manuscript: Considering the clearer images about the selective deposition of Au on MoS_x surface in Fig. R1, the above HRTEM results in Figure R1 was chosen as the typical images to replace the corresponding results in Fig. 2 in our revised manuscript. The corresponding description was added to the revised manuscript.

Comments 2: Please provide the production rate of H₂O₂ of all samples as a plot of time, for longer hours. As produced H₂O₂ accumulates over time, Au will participate in the catalytic degradation of H₂O₂, and the formation of Au^{δ+} sites might play a negative role, in this context. Also, in Fig.3(c), please provide the full data for all samples (1~4).

Author Reply 2: According to the reviewer's suggestion, we performed an extended H₂O₂-production test with longer irradiation hours and the results are shown in Fig. R3. It is found that both TiO₂/MoS_x and TiO₂/Au photocatalysts exhibit a noticeable decrease of H₂O₂ concentration after 4-hour light irradiation, and the TiO₂/Au photocatalyst shows a faster H₂O₂-degradation rate than the TiO₂/MoS_x (Fig. R3-a). In contrast, the TiO₂/MoS_x-Au photocatalyst still maintains a positive H₂O₂-production rate even after 4 hours of irradiation, suggesting the excellent promotion of H₂O₂ production after the selective deposition of Au nanoparticles. In addition, the slight decrease of H₂O₂ concentration after 6-hour illumination can be ascribed to the

insufficient O₂ (Fig. R3-b) and the Au-induced H₂O₂ decomposition. To further confirm the effect of Au nanoparticles on the H₂O₂-decomposition rate, the H₂O₂-decomposition experiments of TiO₂, TiO₂/Au and TiO₂/MoS_x-Au were performed and the results are shown in Fig. R4. Doubtlessly, compared with TiO₂, the presence of Au sites in TiO₂/Au can greatly inhibit the rapid decomposition of H₂O₂. With further introduction of MoS_x, the resulting TiO₂/MoS_x-Au exhibits a further decrease of the H₂O₂-decomposition rate, resulting in an improved photocatalytic H₂O₂ production.

Fig. R3 (a) Time-dependent H₂O₂-evolution activity of TiO₂, TiO₂/MoS_x, TiO₂/Au and TiO₂/MoS_x-Au samples. (b) The enhanced photocatalytic H₂O₂ concentration in the TiO₂/MoS_x-Au suspension after the further introduction of O₂.

Fig. R4 Photocatalytic H₂O₂-degradation experiments of TiO₂, TiO₂/Au and TiO₂/MoS_x-Au samples under irradiation and oxygen-free condition.

Our modification to the manuscript: According to the reviewer's suggestion, the full data for both TiO₂ and TiO₂/MoS_x-Au are presented in Supplementary Fig. 7, which is also shown as follows.

Supplementary Figure 7. Time-dependent H₂O₂-evolution activity of TiO₂ and TiO₂/MoS_x-Au in the O₂-saturated and air condition.

Comments 3: ICP results indicate that a relatively small amount of MoS_x is present

compared to Au, but in Fig.4(e, f), a small charge density change in Mo and S atoms induce a relatively huge charge density change in Au. What is the reason?

Author Reply 3: We are very grateful for the reviewer's professional comment. After careful re-examination of the original data, we found that the Bader charge value of Au in the MoS_x-Au should be 0.07 instead of 0.74. Therefore, the charge density-distribution in Fig. 4e has been revised in our revised manuscript, which is also shown as follows. In Fig. 4f and Supplementary Fig. 10, it was found that compared with the TiO₂/Au, a clear shift of binding energy ($\Delta = 0.4$ eV) to a higher value was observed for Au 4f in the TiO₂/MoS_x-Au, while the XPS peaks of S 2p and Mo 3d shift to lower values ($\Delta = 0.6$ eV for S 2p, $\Delta = 0.3$ eV for Mo 3d) than that of TiO₂/MoS_x. According to the widely reported results¹⁻³, the shifting value of binding energy for Au element in different composite materials is usually in the range of 0.2-0.7 eV. Obviously, our present result about binding energy of Au nanoparticles is reasonable, in good agreement with the above reported results.

Fig. 4e The charge density distributions of MoS_x, MoS_x-Au, and Au.

References:

1. Li, P. et al. Ultrathin porous g-C₃N₄ nanosheets modified with AuCu alloy nanoparticles and C-C coupling photothermal catalytic reduction of CO₂ to ethanol. *Appl. Catal. B Environ.* **266**, 118618 (2020).
2. Shi, H. et al. Selective modification of ultra-thin g-C₃N₄ nanosheets on the (110) facet of Au/BiVO₄ for boosting photocatalytic H₂O₂ production. *Appl. Catal. B Environ.* **297**, 120414 (2021).
3. Gao, D. et al. Optimizing atomic hydrogen desorption of sulfur-rich NiS_{1+x} cocatalyst for boosting photocatalytic H₂ evolution. *Adv. Mater.* **34**, 2108475 (2022).

Our modification to the manuscript: The above Fig. 4e has been added into our revised manuscript.

Comments 4: In accordance with comment 3, for the total charge to be preserved, there might be some electronic structural changes in TiO₂. Please suggest further details about the electronic states of TiO₂, including Ti 2p XPS spectra. Presence of TiO₂ is generally overlooked throughout the manuscript.

Author Reply 4: We appreciate the reviewer's professional comment. According to the reviewer's suggestion, we have conducted an analysis of the Ti 2p spectra for all samples, which are presented in Supplementary Fig. 11. It is found that after the deposition of MoS_x, Au and MoS_x-Au, the resulting samples shows different electronic structure. The binding energies of Ti 2p in TiO₂/Au ($\Delta = 0.06$ eV), TiO₂/MoS_x ($\Delta = 0.13$

eV) and TiO₂/MoS_x-Au ($\Delta = 0.22$ eV) shift to higher values compared to the bare TiO₂, suggesting the free-electron transfer from TiO₂ to Au, MoS_x and MoS_x-Au cocatalysts, respectively.

Supplementary Figure 11. High-resolution XPS spectra of Ti 2p in TiO₂, TiO₂/MoS_x, TiO₂/Au and TiO₂/MoS_x-Au.

Our modification to the manuscript: The above Supplementary Fig. 11 has been added into our Supporting Information, and the corresponding description was added in the revised manuscript.

Comments 5: Does this electron discharging strategy work for other non-plasmonic metals such as Pd? This could not only prove the versatility of the suggested charge redistribution strategy, but also separate the possible effect of surface plasmon resonance.

Author Reply 5: We appreciate the reviewer's comment. The present electron discharging strategy can also be used to other non-plasmonic metals. The main principle about the electron discharging strategy is based on the difference of work function for

different materials. It is well known that Pd has a strong adsorption for O₂ molecular, which often leads to sluggish dynamics and poor selectivity for H₂O₂ production. Therefore, in our another work, we are trying to regulate the electronic structure of Pd by charging its orbital electron via coupling with RuSe₂. The primary aim is to increase the occupancy of antibonding orbitals in Pd-O_{ads}, thereby weakening the O₂ adsorption and achieving efficient H₂O₂ production. Photocatalytic H₂O₂-production test demonstrates that the resulting RuSe₂/Pd-modified TiO₂ shows a greatly enhanced photocatalytic performance. Therefore, we believed that the present charge redistribution strategy exhibits good versatility for the reasonable design of highly efficient photocatalytic materials in various applications.

Comments 6: In Table S2, please also add some references that use Au metals as cocatalysts (preferably more than 5). Some are also mentioned in the introduction section.

Author Reply 6: According to the reviewer's suggestion, some important references about metallic Au cocatalysts for photocatalytic H₂O₂ production have been added in Table S2, which is also shown as follows. Obviously, compared with the widely reported photocatalytic materials (Table S2), the present TiO₂/MoS_x-Au significantly shows a higher photocatalytic H₂O₂-production activity (30.44 mmol g⁻¹ h⁻¹) than most of reported photocatalysts, which indicates that the electron discharging strategy offers a promising approach to enhance O₂ adsorption for efficient photocatalytic H₂O₂ production.

Supplementary Table S2. Comparison of various photocatalytic materials and their corresponding H₂O₂-production rates

Catalysts	Reactant solution	Light source	Rate of H ₂ O ₂ production (mmol g ⁻¹ h ⁻¹)	Ref.
PEI/C ₃ N ₄	H ₂ O	AM 1.5	0.2081	(1) ¹
PCN-NaCA-2	3.5% glycerol	AM 1.5	18.7	(2) ²
Ni ₄ %/O _{0.2} tCN	10% ethanol	300 W Xe lamp $\lambda > 420$ nm	2.464	(3) ³
AKMT	10% ethanol	300 W Xe lamp $\lambda > 420$ nm	2.733	(4) ⁴
5Cv@g-C ₃ N ₄	10% ethanol	300 W Xe lamp $\lambda > 420$ nm	7.01	(5) ⁵
CoPc-BTM-COF	10% ethanol	300 W Xe lamp $\lambda > 400$ nm	2.096	(6) ⁶
TC/g-CN/BOC	5% IPA	300 W Xe lamp	1.275	(7) ⁷
ZnO/WO ₃	10% ethanol	300 W Xe lamp	6.788	(8) ⁸
TiO ₂ @NSG	10% ethanol	300 W Xe lamp	1.746	(9) ⁹
Zn ₃ In ₂ S ₆	Acetonitrile 25 mm THIQs	300 W Xe lamp $\lambda > 400$ nm	66.4	(10) ¹⁰
Au _{0.1} Ag _{0.4} /TiO ₂	4% ethanol	450 W Hg lamp $\lambda > 280$ nm	3.4	(11) ¹¹
Cu@Au/BiVO ₄	5% methanol	12.5 mW LED lamp $\lambda = 420$ nm	0.118	(12) ¹²
C ₃ N ₄ -Au/BiVO ₄	0.2 M citrate buffer solution	50 mW LED lamp $\lambda = 420$ nm	0.676	(13) ¹³
BiVO ₄ /AuPd	0.2 M citrate buffer solution	20 mW LED lamp $\lambda = 420$ nm	1.145	(14) ¹⁴
C ₃ N ₄ /Au	10% ethanol	300 W Xe lamp $\lambda > 420$ nm	0.017	(15) ¹⁵
WO ₃ /Au	4% methanol	400 W metal halide lamp $\lambda > 420$ nm	0.111	(16) ¹⁶
OVs-BiOBr-Au	5% formic acid	300 W Xe lamp $\lambda > 420$ nm	0.127	(17) ¹⁷
TiO ₂ /MoS _x -Au	10% ethanol	300 W Xe lamp	30.44	This work

Supplementary References

1. Zeng, X. et al. Simultaneously tuning charge separation and oxygen reduction pathway on graphitic carbon nitride by polyethylenimine for boosted photocatalytic hydrogen peroxide production. *ACS Catal.* **10**, 3697-3706 (2020).
2. Zhao, Y. et al. Mechanistic analysis of multiple processes controlling solar-driven H₂O₂ synthesis using engineered polymeric carbon nitride. *Nat. Commun.* **12**, 3701 (2021).
3. Du, R. et al. Controlled oxygen doping in highly dispersed Ni-loaded g-C₃N₄ nanotubes for efficient photocatalytic H₂O₂ production. *Chem. Eng. J.* **441**, 135999 (2022).
4. Zhang, P. et al. Heteroatom dopants promote two-electron O₂ reduction for photocatalytic production of H₂O₂ on polymeric carbon nitride. *Angew. Chem. Int. Ed.* **59**, 16209-16217 (2020).
5. Chen, L. et al. Simultaneously tuning band structure and oxygen reduction pathway toward high-efficient photocatalytic hydrogen peroxide production using cyano-rich graphitic carbon nitride. *Adv. Funct. Mater.* **31**, 2105731 (2021).
6. Zhi, Q. et al. Piperazine-linked metalphthalocyanine frameworks for highly efficient visible-light-driven H₂O₂ photosynthesis. *J. Am. Chem. Soc.* **144**, 21328-21336 (2022).
7. Yang, Q., Li, R., Wei, S., Yang, R. Schottky functionalized Z-scheme heterojunction photocatalyst Ti₂C₃/g-C₃N₄/BiOCl: Efficient photocatalytic

- H₂O₂ production via two-channel pathway. *Appl. Surf. Sci.* **572**, 151525 (2022).
8. Jiang, Z. et al. S-scheme ZnO/WO₃ heterojunction photocatalyst for efficient H₂O₂ production. *J. Mater. Sci. Technol.* **124**, 193-201 (2022).
 9. Yang, Y. et al. In-situ grown N, S co-doped graphene on TiO₂ fiber for artificial photosynthesis of H₂O₂ and mechanism study. *Appl. Catal. B Environ.* **317**, 121788 (2022).
 10. Luo, J. et al. Photoredox-promoted Co-production of dihydroisoquinoline and H₂O₂ over defective Zn₃In₂S₆. *Adv. Mater.* **35**, 2210110 (2023).
 11. Tsukamoto, D. et al. Photocatalytic H₂O₂ production from ethanol/O₂ system using TiO₂ loaded with Au-Ag bimetallic alloy nanoparticles. *ACS Catal.* **2**, 599-603 (2012).
 12. Wang, K. et al. BiVO₄ Microparticles decorated with Cu@Au core-shell nanostructures for photocatalytic H₂O₂ production. *ACS Appl. Nano Mater.* **4**, 13158-13166 (2021).
 13. Shi, H. et al. Selective modification of ultra-thin g-C₃N₄ nanosheets on the (110) facet of Au/BiVO₄ for boosting photocatalytic H₂O₂ production. *Appl. Catal. B Environ.* **297**, 120414 (2021).
 14. Shi, H. et al. Mass-transfer control for selective deposition of well-dispersed AuPd cocatalysts to boost photocatalytic H₂O₂ production of BiVO₄. *Chem. Eng. J.* **443**, 136429 (2022).
 15. Zuo, G. et al. Finely dispersed Au nanoparticles on graphitic carbon nitride as highly active photocatalyst for hydrogen peroxide production. *Catal. Commun.*

123, 69-72 (2019).

16. Wang, Y. et al. Efficient production of H₂O₂ on Au/WO₃ under visible light and the influencing factors. *Appl. Catal. B Environ.* **284**, 119691 (2021).
17. An, R. et al. Decoration of Au NPs on hollow structured BiOBr with surface oxygen vacancies for enhanced visible light photocatalytic H₂O₂ evolution. *J. Solid State Chem.* **306**, 122722 (2022).

Reviewer #2 (Remarks to the Author):

This work by Xidong Zhang et al. reports on photocatalytic production of H₂O₂ with a rate that is very high indeed compared to the recent literatures. The authors attribute this success partial charge transfer from Au nanoparticles (at which oxygen reaction is supposed to take place) to underlying MoS_x due to the difference of their work functions. The preparation of the catalyst, its characterization, experimental conditions, and computational modelling details are described in sufficient detail. The work functions from the DFT calculations are in good agreement with published experimental values.

Comments 1: Concerning the results obtained at different catalysts, there is some confusion if one compares Figure 3a and the statements at different places of the text, in which the less efficient catalyst compositions are presented in different order:

87-88: The synthesized TiO₂/MoS_x-Au photocatalyst achieves a boosted H₂O₂-production rate of 30.44 mmol g⁻¹ h⁻¹, which is 25.4 and 1.3 times higher than that of TiO₂ and TiO₂/Au, respectively.

167-169: the optimal photocatalytic H₂O₂ yield in TiO₂/MoS_x-Au reached to 30.44 mmol g⁻¹ h⁻¹ (Fig. 3a and Supplementary Fig. 6), which is 25.4 and 1.3 times higher than that of TiO₂/Au and TiO₂/MoS_x, respectively.

Author Reply 1: We are very grateful for the reviewer's comment. The sample names of TiO₂/Au and TiO₂/MoS_x in the "the optimal photocatalytic H₂O₂ yield in TiO₂/MoS_x-Au reached to 30.44 mmol g⁻¹ h⁻¹ (Fig. 3a and Supplementary Fig. 6), which is 25.4 and 1.3 times higher than that of TiO₂/Au and TiO₂/MoS_x, respectively" should be TiO₂ and

TiO₂/Au, respectively. Herein, we have revised it in our manuscript, which is also shown as follows.

“The optimal photocatalytic H₂O₂ yield in TiO₂/MoS_x-Au reached 30.44 mmol g⁻¹ h⁻¹ (Fig. 3a and Supplementary Fig. 6), which is 25.4 and 1.3 times higher than that of TiO₂ and TiO₂/Au, respectively.”

Comments 2: Supposing Fig. 3a is correct, the effect of MoS_x underlayer on the increase of Au activity is relatively weak (30%), while the authors see in it the main message of their work. On the other hand, Au/TiO₂ is already very active, compared to the literature. An important result presented in Supplementary Figure 6 but not discussed is virtual independence of rate on Au content changing 5-fold (from 1 to 5%). How is it possible that O₂ is supposed to be chemisorbed on Au but increased amount (surface?) of Au does not produce more H₂O₂. At the same time, the authors are sure that Au is at the surface of MoS_x:

Author Reply 2: We appreciate the reviewer’s professional comment. As mentioned by the reviewer, the present TiO₂/Au shows a high photocatalytic H₂O₂-production performance (24.22 mmol g⁻¹ h⁻¹), which is surpassing most of the reported literatures (Table S2, Supplementary Materials). With the further introduction of MoS_x between Au and TiO₂, the resulting TiO₂/MoS_x-Au exhibits an improved H₂O₂-production activity of 30.44 mmol g⁻¹ h⁻¹. Herein, the increased value of 30% is corresponding to a H₂O₂-production rates of over 6 mmol g⁻¹ h⁻¹, suggesting a significant breakthrough in photocatalytic H₂O₂-production field.

In our Supplementary Fig. 6, it is found that with the increase of Au content from 1 to 5%, the resulting TiO₂/MoS_x-Au shows an increased H₂O₂-production rate from ca. 25 to 30 mmol g⁻¹ h⁻¹ and then further decrease to ca. 27 mmol g⁻¹ h⁻¹. In fact, compared with the TiO₂/MoS_x-Au-1%, an increase value of ca. 5 mmol g⁻¹ h⁻¹ was observed for TiO₂/MoS_x-Au-3%, which is a relatively large value in photocatalytic H₂O₂-production field. On the other hand, it is well known that the photocatalytic H₂O₂-production activity is strongly dependence on the simultaneous and effective consumption of photogenerated electrons and holes. Considering the rapid H₂O₂ production by photogenerated electrons on Au active sites, the relatively limited enhancement of H₂O₂-production activity for TiO₂/MoS_x-Au system can possibly be caused by the oxidation rate of photogenerated holes. Further investigations should be confirmed in our further works. In addition, an excessive amount (5%) of Au can cause a light-shielding effect, leading to a decrease in H₂O₂ production. Therefore, the H₂O₂ production does not increase proportionally with the Au surface area.

Our modification to the manuscript: The corresponding discussion about the Supplementary Fig. 6 was provided in our revised manuscript.

Comments 3: 156-158: the contents of Mo, S, and Au elements in the TiO₂/MoS_x-Au photocatalyst are 0.62, 0.8, and 3.34 wt%, respectively. Doubtlessly, these above results collectively support the selective deposition of Au on the MoS_x surface in the resulting TiO₂/MoS_x-Au photocatalyst. This needs to be discussed and clarified.

Author Reply 3: According to the reviewer's suggestion, the above points have been further discussed and clarified, which is as shown as follows.

“According to the inductively coupled plasma optical emission spectrometry (ICP-OES) results (Supplementary Table S1), the contents of Mo, S, and Au elements in the TiO₂/MoS_x-Au photocatalyst are 0.62, 0.8, and 3.34 wt%, respectively, indicating the presence of MoS_x and Au in the TiO₂/MoS_x-Au system. Doubtlessly, the above results (HRTEM, UV-Vis, and ICP-OES) further support the selective deposition of Au on the MoS_x surface in the resulting TiO₂/MoS_x-Au photocatalyst.”

To further determine whether all Au nanoparticles are selectively deposited on the MoS_x surface, we perform the further HRTEM analysis for different regions of TiO₂/MoS_x-Au sample, which can provide direct and visual evidence of the selective deposition for Au nanoparticles. A typical HRTEM result of the TiO₂/MoS_x-Au sample is shown in Fig. R1. It is found that there is some amorphous structure between the Au and TiO₂ nanoparticles in Figure R1c, which can be attributed to the amorphous MoS_x, clearly suggesting the excellent coupling of Au on MoS_x surface. In Fig. R1e, in addition to two larger Au nanoparticles (ca. 5 nm), two smaller Au can also be observed on a TiO₂ particle surface. Their corresponding EDS mapping images show that the four Au nanoparticles exhibit uniform and same distribution with the Mo and S elements on the TiO₂ particles, unequivocally indicating that all the Au nanoparticles are selectively deposited on the MoS_x surface, which is independent of Au particle size.

Fig. R1 (a, b) TEM, (c) HRTEM, (d, e) HAADF-STEM, and EDS mapping images of TiO₂/MoS_x-Au.

Our modification to the manuscript: The above description was modified in our revised manuscript.

Comments 4: In all, there is vast literature on H₂O₂ photocatalytic production at TiO₂/Au; which merits a better discussion than provided in the Introduction. The roles of Au and MoS_x underlayer remain not quite clear. I invite the authors to reconsider the interpretation of their results and reconsider the work for publication afterwards.

Author Reply 4: We appreciate the reviewer's professional comment. There are numerous research works about the photocatalytic H₂O₂-production by using TiO₂/Au

systems. Notably, these studies have primarily focused on effectively separating photogenerated electron-hole pairs by using gold as a co-catalyst^{1,2}. Notably, the fundamental understanding of the electronic structure of Au active sites on the O₂ adsorption and H₂O₂-production activity is still unclear. In our work, an ingenious strategy about releasing d-orbital electrons of Au cocatalyst is proposed to strengthen O₂ adsorption for improving H₂O₂-production kinetics.

To improve the O₂ adsorption on the Au active sites, we introduced MoS_x to induce selective deposition of Au onto the MoS_x surface by the self-assembly of S-Au, and deeply investigated the impact of introducing MoS_x on the electronic structure of Au and its photocatalytic H₂O₂-production activity through experiments and theoretical calculations. As a result, the free electrons would migrate inevitably from Au nanoparticles to MoS_x due to the difference of work functions between the MoS_x (5.8 eV) and Au (5.2 eV), leading to the formation of electron-deficient Au^{δ+} active sites. Consequently, the introduction of MoS_x mediator can effectively induce the d-orbital electron transfer of Au cocatalyst, which can be further verified by local charge density difference, Bader charge calculation, and XPS. Moreover, O₂ adsorption energy, crystal orbital Hamilton population (COHP), and partial density of states (PDOS) calculations results indicate that the formation of electron-deficient Au^{δ+} active sites effectively decrease the antibonding-orbital occupancy of the Au-O_{ads}. Consequently, the adsorption strength of Au^{δ+} active sites towards O₂ is significantly enhanced, resulting in significant improvements in both selectivity and activity of photocatalytic H₂O₂ production. Therefore, the presence of MoS_x not only induces selective deposition of

Au onto the MoS_x surface but also modulates the electronic structure of Au, leading to the enhancement of O₂ adsorption at the Au sites for efficient H₂O₂ production.

References:

1. Kim, K. et al. Solid-phase photocatalysts: Physical vapor deposition of Au nanoislands on porous TiO₂ films for millimolar H₂O₂ production within a few minutes. *ACS Catal.* **9**, 9206-9211 (2019).
2. Tsukamoto, D. et al. Photocatalytic H₂O₂ production from ethanol/O₂ system using TiO₂ loaded with Au-Ag bimetallic alloy nanoparticles. *ACS Catal.* **2**, 599-603 (2012).

Our modification to the manuscript: The advances about Au-loaded photocatalysts for H₂O₂ production were further discussed in our Introduction, which is also shown as follows.

“After the rapid capture of photogenerated electrons on Au surface, the adsorbed O₂ can be reduced to produce H₂O₂ via a two-step single-electron or one-step two-electron ORR processes³⁰⁻³².”

References:

30. Zuo, G. et al. Finely dispersed Au nanoparticles on graphitic carbon nitride as highly active photocatalyst for hydrogen peroxide production. *Catal. Commun.* **123**,

69-72 (2019).

31. Kim, K. et al. Solid-phase photocatalysts: Physical vapor deposition of Au nanoislands on porous TiO₂ films for millimolar H₂O₂ production within a few minutes. *ACS Catal.* **9**, 9206-9211 (2019).

32. An, R. et al. Decoration of Au NPs on hollow structured BiOBr with surface oxygen vacancies for enhanced visible light photocatalytic H₂O₂ evolution. *J. Solid State Chem.* **306**, 122722 (2022).

Reviewer #3 (Remarks to the Author):

The manuscript describes the synthesis of a TiO₂/MoS_x-Au composite and its use for photocatalytic H₂O₂ generation with ethanol as an electron donor. The authors claimed enhanced H₂O₂ generation by the creation of the three component systems. However, the manuscript contains several problems at the present form. Followings should be considered.

Comments 1: TiO₂/MoS_x is a well-known photocatalyst for enhancing charge separation (<https://doi.org/10.1016/j.apcatb.2021.120878>). TiO₂/Au is a well-known photocatalyst for H₂O₂ generation (<https://doi.org/10.1021/ja102651g>). Moreover, MoS_x/Au has also already been reported to behaves as a good photocatalyst for H₂O₂ generation (<https://doi.org/10.1016/j.jcat.2019.06.015>). Based on this situation, the present photocatalyst is just the combination of these reported materials. I therefore must say that the novelty of this manuscript is insufficient.

Author Reply 1: We appreciate the reviewer's critical comments. The above works have been carefully read and their corresponding novelty are shown as follows, respectively.

The first work (<https://doi.org/10.1016/j.apcatb.2021.120878>) is focused on constructing a dual-defect (the presence of both O vacancies and S vacancies) heterojunction system of TiO₂ hierarchical microspheres with oxygen vacancies modified with ultrathin MoS_{2-x} nanosheets for simultaneously degrading pollutants and evolving hydrogen. In treating the simulated pharmaceutical wastewater, MoS_{2-x}@TiO₂-OV is capable of purifying various refractory contaminants, with the highest

H₂ evolution rate of 41.59 $\mu\text{mol g}^{-1} \text{h}^{-1}$ during enrofloxacin degradation. While treating the simulated coking wastewater, the catalyst achieves a H₂ evolution rate of 102.72 $\mu\text{mol g}^{-1} \text{h}^{-1}$ and a mineralization rate of 50%. Computational studies suggest that the dual-defect is superior for the adsorption of H* and producing $\cdot\text{OH}$ ('dual-defect boosted dual-function'). Also, the dual-defect sites significantly boosted the charge-carrier separation and transfer efficiencies.

The second work (<https://doi.org/10.1021/ja102651g>) is focused on the impact of the Au-nanoparticle size on the photocatalytic H₂O₂-production performance of TiO₂-Au and provides insight into the relationship between Au-nanoparticle sizes and rate constants of formation (K_f) and decomposition (K_d). The results demonstrate that TiO₂-Au exhibits a higher photocatalytic activity for H₂O₂-production than TiO₂ and TiO₂/Pt, which can be attributed to the effective separation of photogenerated carriers, as well as the high formation rate and low degradation rate of H₂O₂.

In the third work (<https://doi.org/10.1016/j.jcat.2019.06.015>), Au-modified MoS₂ nanosheets (Au@MoS₂) for photocatalytic production of H₂O₂ were prepared via a simple pathway including the deposition–reduction and immobilization process by using a low dosage of an Au source. Au modification brought out the low recombination rate of e⁻-h⁺ pairs, long lifetime of electrons, and more negative flat band potential for MoS₂. Au@MoS₂ achieved efficient photocatalytic production of H₂O₂ from H₂O and air in the absence of pure O₂ and organic electron donors. In addition, Mn²⁺ as the active center for Fenton-like reactions was doped in MoS₂ nanosheets before Au⁰ modification in order to use the photogenerated H₂O₂ in situ. Accordingly, a novel in situ Fenton

process was proposed, and obtained significant degradation efficiencies for rhodamine B and methylene blue dyes, depending on the H₂O₂ productivity.

Obviously, the above three works share a common feature in material design, which is the coupling of an additional co-catalyst to achieve effective separation of photogenerated carriers, resulting in efficient oxidation and reduction reactions. However, in our present work, the main novelty of this manuscript is focused on the selective deposition of Au nanoparticles on the MoS_x surface to manipulate the electronic structure of Au active sites to increase its O₂ adsorption for efficient H₂O₂ production. As highlighted in the introduction, the adsorption of O₂ on active sites is one of the most important progresses, which is essential for the formation of intermediate *OOH and its subsequent conversion into H₂O₂. However, current cocatalysts usually suffer from a mismatch in electronic configuration between the active site and the adsorbed O₂, leading to either excessively strong or weak O₂ adsorption and limited H₂O₂-production rate. Despite exhibiting high photocatalytic H₂O₂-production activity of Au cocatalyst, its weak adsorption characteristics toward O₂ hinder the formation of *OOH and subsequent H₂O₂ production. Moreover, the fundamental understanding about the dependence of the electronic structure of Au on the Au-O_{ads} bonds and H₂O₂-production activity is still unclear, leading to an ambiguous mechanism for optimizing the Au electronic structure and Au-O_{ads} bonding strength. In this case, based on the molecular orbital theory, we introduced MoS_x into the well-known TiO₂/Au system with the selective deposition of Au on MoS_x surface, and deeply

investigated the effect of MoS_x on the electronic structure of Au and its photocatalytic H₂O₂-production activity through experiments and theoretical calculations. This is an innovative concept using an efficient mediator to optimize the electronic structure of catalytic active sites for realizing their highly effective catalytic reactions. The present strategy is very helpful for the rational design of surface structure for efficient artificial photosynthesis. Based on the aforementioned analysis, we believe that this work exhibits sufficient novelty and represents a significant contribution to the catalytic field.

Comments 2: The biggest problem is Figure 3a. The activity of TiO₂/MoS_x-Au (4) is just a sum of TiO₂/MoS_x (2) and TiO₂/Au (3). This means that “effect of three component system is negligible.” The authors should reconsider this situation, which is the part that decisively affects the importance of this paper.

Author Reply 2: We appreciate the reviewer’s critical comments. In this work, it should be clarified that the H₂O₂-production activity of TiO₂/MoS_x-Au cannot be evaluated as the sum of TiO₂-Au and TiO₂-MoS_x. In the photocatalytic process, O₂ is initially adsorbed onto the active sites, which subsequently capture the photogenerated electrons to produce H₂O₂. It is worth noting that the active sites in the TiO₂/MoS_x and TiO₂/Au photocatalysts are located on the Mo and Au sites, respectively. However, in the TiO₂/MoS_x-Au system, Au nanoparticles are in-situ reduced and covered on the MoS_x surface by the self-assembling of S-Au, as well confirmed by the HRTEM images in Fig. 2. Hence, we believe that only the Au active sites participate in the H₂O₂ production instead of both Mo and Au sites. Moreover, the enhancement of

photocatalytic H₂O₂-production performance over TiO₂/MoS_x-Au can be attributed to the introduction of MoS_x mediator, which effectively modulates the electronic structure of Au cocatalyst, thereby decreasing the antibonding-orbital occupancy of Au-O_{ads} for enhancing O₂ adsorption and realizing excellent H₂O₂-production performance.

Comments 3: The basic problem of this manuscript is that the system requires ethanol as an electron donor. Several photocatalytic systems using sacrificial electron donors have been reported; many of them shows much higher activity than the present system, although the authors did not cite these papers at al.

Author Reply 3: We appreciate the reviewer's critical comments. Currently, solar-driven H₂O₂ production are mainly focused on the two different experimental conditions, such as pure O₂/H₂O and O₂/sacrificial agent systems. The photocatalytic H₂O₂ production from the O₂-H₂O system is gradually becoming a research hotspot, which can use photogenerated electrons and holes to synthesize H₂O₂ by the O₂ reduction reaction (ORR) and H₂O oxidation reaction (WOR), respectively. However, the photocatalytic H₂O₂-production yield in the O₂-H₂O system is usually far lower than that in the presence of a sacrificial agent.¹⁻³ The main reasons are summarized as follows. First, the kinetically sluggish four-electron oxidation of H₂O (WOR, ~ s, H₂O + 4h⁺ → O₂ + 4H⁺) cannot effectively serves as the proton source for the faster ORR (μs ~ ms), leading to the serious inadequacies in proton supply for ORR⁴. Secondly, the stepwise single-electron WOR process (·OH + ·OH → H₂O₂) to generate H₂O₂ requires a high concentration of ·OH (+ 2.73 V vs NHE at pH=0), which severely limits the

H₂O₂ yield⁵. Finally, the direct formation of H₂O₂ via two-electron WOR ($2\text{H}_2\text{O} + 2\text{h}^+ \rightarrow \text{H}_2\text{O}_2 + 2\text{H}^+$) by photogenerated holes is still a challenge⁶. This is due to the thermodynamic preference for the four-electron water oxidation to produce O₂ (+1.23 eV vs NHE at pH=0) over the two-electron WOR to form H₂O₂ (+1.78 eV vs NHE at pH=0)⁷. Therefore, based on the above analysis, photocatalytic materials usually exhibit a limited H₂O₂-production activity in the O₂-H₂O system.

To greatly improve the H₂O₂-production activity, various sacrificial agents have been widely introduced, which is recognized as an effective mean to promote the O₂-reduction reaction for the H₂O₂ production. Herein, we use ethanol as an electron donor to evaluate the photocatalytic H₂O₂-production rate of the TiO₂/MoS_x-Au. Obviously, compared with the widely reported photocatalytic materials (Table S2), the present TiO₂/MoS_x-Au significantly shows a higher photocatalytic H₂O₂-production activity (30.44 mmol g⁻¹ h⁻¹) than most of reported photocatalysts. More importantly, the main novelty of this manuscript is focused on the selective deposition of Au nanoparticles on the MoS_x surface to manipulate the electronic structure of Au active sites to increase its O₂ adsorption for efficient H₂O₂ production. Density functional theory calculations and XPS analysis confirm that the MoS_x mediator can induce the formation of electron-deficient Au^{δ+} sites by releasing Au d orbital electrons to MoS_x, thereby decreasing the antibonding-orbital occupancy of Au-O_{ads} for enhancing O₂ adsorption. Femtosecond transient absorption spectra and in situ KPFM further demonstrate the effective transfer of photogenerated electrons in the TiO₂/MoS_x-Au photocatalyst.

In addition, some important references about highly efficient H₂O₂ production are

summarized in Table S2 in Supporting Information, which is also shown as follows.

References:

1. Li, Y. et al. Edge engineering of carbon nitride for enhanced sacrificial agent-free photocatalytic H₂O₂ evolution. *Chem. Eng. J.* **282**, 119333 (2023).
2. Yang, C. et al. Calcination-regulated microstructures of donor-acceptor polymers towards enhanced and stable photocatalytic H₂O₂ production in pure water. *Angew. Chem. Int. Ed.* **134**, e202208438 (2022).
3. Zhang, X. et al. C₃N₄/PDA S-Scheme heterojunction with enhanced photocatalytic H₂O₂ production performance and its mechanism. *Adv. Sustainable Syst.* **7**, 2200113 (2023).
4. Sheng, B. et al. Proton reservoirs in polymer photocatalysts for superior H₂O₂ photosynthesis. *Energy Environ. Sci.* **16**, 4612-4619 (2023).
5. Cheng, H. et al. Reaction pathways toward sustainable photosynthesis of hydrogen peroxide by polymer photocatalysts. *Chem. Mater.* **34**, 4259-4273 (2022).
6. Chen, L. et al. Acetylene and diacetylene functionalized covalent triazine frameworks as metal-free photocatalysts for hydrogen peroxide production: A new two-electron water oxidation pathway. *Adv. Mater.* **32**, 1904433 (2020).
7. Liu, J. et al. Metal-free efficient photocatalyst for stable visible water splitting via a two-electron pathway. *Science* **347**, 970-974 (2016).

Supplementary Table S2. Comparison of various photocatalytic materials and their corresponding H₂O₂-production rates

Catalysts	Reactant solution	Light source	Rate of H ₂ O ₂ production (mmol g ⁻¹ h ⁻¹)	Ref.
PEI/C ₃ N ₄	H ₂ O	AM 1.5	0.2081	(1) ¹
PCN-NaCA-2	3.5% glycerol	AM 1.5	18.7	(2) ²
Ni ₄ %/O _{0.2} tCN	10% ethanol	300 W Xe lamp λ>420 nm	2.464	(3) ³
AKMT	10% ethanol	300 W Xe lamp λ>420 nm	2.733	(4) ⁴
5Cv@g-C ₃ N ₄	10% ethanol	300 W Xe lamp λ>420 nm	7.01	(5) ⁵
CoPc-BTM-COF	10% ethanol	300 W Xe lamp λ>400 nm	2.096	(6) ⁶
TC/g-CN/BOC	5% IPA	300 W Xe lamp	1.275	(7) ⁷
ZnO/WO ₃	10% ethanol	300 W Xe lamp	6.788	(8) ⁸
TiO ₂ @NSG	10% ethanol	300 W Xe lamp	1.746	(9) ⁹
Zn ₃ In ₂ S ₆	Acetonitrile 25 mm THIQs	300 W Xe lamp λ>400 nm	66.4	(10) ¹⁰
Au _{0.1} Ag _{0.4} /TiO ₂	4% ethanol	450 W Hg lamp λ>280 nm	3.4	(11) ¹¹
Cu@Au/BiVO ₄	5% methanol	12.5 mW LED lamp λ=420 nm	0.118	(12) ¹²
C ₃ N ₄ -Au/BiVO ₄	0.2 M citrate buffer solution	50 mW LED lamp λ=420 nm	0.676	(13) ¹³
BiVO ₄ /AuPd	0.2 M citrate buffer solution	20 mW LED lamp λ=420 nm	1.145	(14) ¹⁴
C ₃ N ₄ /Au	10% ethanol	300 W Xe lamp λ>420 nm	0.017	(15) ¹⁵
WO ₃ /Au	4% methanol	400 W metal halide lamp λ>420 nm	0.111	(16) ¹⁶
OVs-BiOBr-Au	5% formic acid	300 W Xe lamp λ>420 nm	0.127	(17) ¹⁷
TiO ₂ /MoS _x -Au	10% ethanol	300 W Xe lamp	30.44	This work

Supplementary References

1. Zeng, X. et al. Simultaneously tuning charge separation and oxygen reduction pathway on graphitic carbon nitride by polyethylenimine for boosted photocatalytic hydrogen peroxide production. *ACS Catal.* **10**, 3697-3706 (2020).
2. Zhao, Y. et al. Mechanistic analysis of multiple processes controlling solar-driven H₂O₂ synthesis using engineered polymeric carbon nitride. *Nat. Commun.* **12**, 3701 (2021).
3. Du, R. et al. Controlled oxygen doping in highly dispersed Ni-loaded g-C₃N₄ nanotubes for efficient photocatalytic H₂O₂ production. *Chem. Eng. J.* **441**, 135999 (2022).
4. Zhang, P. et al. Heteroatom dopants promote two-electron O₂ reduction for photocatalytic production of H₂O₂ on polymeric carbon nitride. *Angew. Chem. Int. Ed.* **59**, 16209-16217 (2020).
5. Chen, L. et al. Simultaneously tuning band structure and oxygen reduction pathway toward high-efficient photocatalytic hydrogen peroxide production using cyano-rich graphitic carbon nitride. *Adv. Funct. Mater.* **31**, 2105731 (2021).
6. Zhi, Q. et al. Piperazine-linked metalphthalocyanine frameworks for highly efficient visible-light-driven H₂O₂ photosynthesis. *J. Am. Chem. Soc.* **144**, 21328-21336 (2022).
7. Yang, Q., Li, R., Wei, S., Yang, R. Schottky functionalized Z-scheme heterojunction photocatalyst Ti₂C₃/g-C₃N₄/BiOCl: Efficient photocatalytic

- H₂O₂ production via two-channel pathway. *Appl. Surf. Sci.* **572**, 151525 (2022).
8. Jiang, Z. et al. S-scheme ZnO/WO₃ heterojunction photocatalyst for efficient H₂O₂ production. *J. Mater. Sci. Technol.* **124**, 193-201 (2022).
 9. Yang, Y. et al. In-situ grown N, S co-doped graphene on TiO₂ fiber for artificial photosynthesis of H₂O₂ and mechanism study. *Appl. Catal. B Environ.* **317**, 121788 (2022).
 10. Luo, J. et al. Photoredox-promoted Co-production of dihydroisoquinoline and H₂O₂ over defective Zn₃In₂S₆. *Adv. Mater.* **35**, 2210110 (2023).
 11. Tsukamoto, D. et al. Photocatalytic H₂O₂ production from ethanol/O₂ system using TiO₂ loaded with Au-Ag bimetallic alloy nanoparticles. *ACS Catal.* **2**, 599-603 (2012).
 12. Wang, K. et al. BiVO₄ Microparticles decorated with Cu@Au core-shell nanostructures for photocatalytic H₂O₂ production. *ACS Appl. Nano Mater.* **4**, 13158-13166 (2021).
 13. Shi, H. et al. Selective modification of ultra-thin g-C₃N₄ nanosheets on the (110) facet of Au/BiVO₄ for boosting photocatalytic H₂O₂ production. *Appl. Catal. B Environ.* **297**, 120414 (2021).
 14. Shi, H. et al. Mass-transfer control for selective deposition of well-dispersed AuPd cocatalysts to boost photocatalytic H₂O₂ production of BiVO₄. *Chem. Eng. J.* **443**, 136429 (2022).
 15. Zuo, G. et al. Finely dispersed Au nanoparticles on graphitic carbon nitride as highly active photocatalyst for hydrogen peroxide production. *Catal. Commun.*

123, 69-72 (2019).

16. Wang, Y. et al. Efficient production of H₂O₂ on Au/WO₃ under visible light and the influencing factors. *Appl. Catal. B Environ.* **284**, 119691 (2021).
17. An, R. et al. Decoration of Au NPs on hollow structured BiOBr with surface oxygen vacancies for enhanced visible light photocatalytic H₂O₂ evolution. *J. Solid State Chem.* **306**, 122722 (2022).

Our modification to the manuscript: Some important references about highly efficient H₂O₂ production are summarized in Table S2 in Supporting Information.

Comment 4: Based on the above, I do not think the novelty and quality of this manuscript justify publication in this journal.

Author Reply 4: We highly appreciate the comments from the reviewer. The above comments are carefully considered and well provided in our revised manuscript.

Thank you very much again for your kind and appropriate comments. We are sure that these comments help improve the quality of our manuscripts significantly.

REVIEWER COMMENTS

Reviewer #1 (Remarks to the Author):

The authors have well revised the manuscript following reviewer comments, and the reviewer believes that the manuscript is now ready for publication.

Reviewer #2 (Remarks to the Author):

It is fortunate that the authors could correct the DFT computed Bader charge on Au in the triple system from +0,74 to +0,07, which makes charge transfer obtained by theory pretty small. In spite of this, XPS binding energy shift of +0,4 eV for Au remains an argument in favor of positively charged Au. However, the choice of the title "Releasing d-orbital electron of Au cocatalyst for improved photocatalytic H₂O₂ production" is now less certain.

Similarly to Reviewer 3 Comment 2, I was and still am concerned with the role of various components of the catalyst. The authors believe that O₂ is adsorbed and reacts only on Au. This would imply that MoS_x is fully covered with Au, given that MoS_x on TiO₂ is a good photocatalyst itself. However, the authors' own representation of Au clusters on MoS_x, as illustrated in Figure 2a (understood: this is just an illustration, not a proof!) is rather correct, and O₂ reaction at MoS_x cannot be excluded.

In all, though appreciating this vast multidisciplinary study, I am not convinced that the results, their interpretation, and significance are at the level required for publication in Nature Communications.

Reviewer #3 (Remarks to the Author):

I was satisfied with the revision.

Responses to the reviewers' comments

Reviewer #1 (Remarks to the Author):

Comments: The authors have well revised the manuscript following reviewer comments, and the reviewer believes that the manuscript is now ready for publication.

Author Reply 1: We are very grateful for the reviewer's comment.

Reviewer #2 (Remarks to the Author):

Comments 1: It is fortunate that the authors could correct the DFT computed Bader charge on Au in the triple system from +0,74 to +0,07, which makes charge transfer obtained by theory pretty small. In spite of this, XPS binding energy shift of +0,4 eV for Au remains an argument in favor of positively charged Au. However, the choice of the title "Releasing d-orbital electron of Au cocatalyst for improved photocatalytic H₂O₂ production" is now less certain.

Author Reply 1: We are very grateful for the reviewer's comment. In this work, it is found that MoS_x mediator can induce the formation of electron-deficient Au^{δ+} sites in the TiO₂/MoS_x-Au photocatalyst to greatly strengthen O₂ adsorption for improving photocatalytic H₂O₂-production activity, which can be well demonstrated by various experimental results, such as work function, local charge density difference, Planar-averaged electron density difference, Bader charge calculation and XPS analysis. First, based on Figs. 4a and b, the work functions (Φ) of MoS_x (001) and Au (111) are calculated to be 5.86 and 5.20 eV, respectively. In this case, when Au is loaded onto

MoS_x surface, the free electrons inevitably migrate from Au nanoparticles to MoS_x, thus inducing the formation of electron-deficient Au^{δ+} sites. Secondly, the local charge density difference and Planar-averaged electron density difference (Figs. 4c and d) suggest that the free electrons are mainly localized on the MoS_x side in the MoS_x-Au cocatalyst to create an electron-enriched region, while positive charges predominantly accumulate on Au atoms, leading to the production of an electron-deficient Au^{δ+} layer. In addition, Bader charge calculation results (Fig. 4e) indicate that the charge densities of S and Mo atoms in MoS_x-Au become more negative compared with pure MoS_x, while that of the Au atoms is slightly increased (+0.07) to produce electron-deficient Au^{δ+} active sites in the MoS_x-Au cocatalyst. Finally, XPS analysis reveals (Fig. 4f) that compared with the TiO₂/Au, a clear shift of binding energy ($\Delta = 0.4$ eV) to a higher value is observed for Au 4f in the TiO₂/MoS_x-Au, while the XPS peaks of S 2p and Mo 3d shift to lower values ($\Delta = 0.6$ eV for S 2p, $\Delta = 0.3$ eV for Mo 3d) than that of TiO₂/MoS_x, indicating the efficient electron transfer from Au to MoS_x. Unquestionably, the above results strongly support that the introduction of MoS_x mediator can effectively modulate the Au electronic structure to induce the formation of electron-deficient Au^{δ+} sites in the MoS_x-Au cocatalyst. Therefore, we believe that the title “Releasing d-orbital electron of Au cocatalyst for improved photocatalytic H₂O₂ production” is reasonable in the present work.

Fig. 4 MoS_x-induced electron-deficient Au^{δ+} formation and mechanism. Calculated average potential profiles of **a** MoS_x and **b** Au. **c** The local charge density difference of MoS_x-Au, where the light yellow and cyan areas represent electron accumulation and depletion, respectively. **d** Planar-averaged electron density difference $\Delta\rho(z)$ in MoS_x-Au. **e** The charge density distributions of MoS_x, MoS_x-Au, and Au. **f** High-resolution XPS spectra of Au 4f in the TiO₂/Au and TiO₂/MoS_x-Au. **g** Schematic illustration about the formation of electron-deficient Au^{δ+} sites in the MoS_x-Au cocatalyst.

Comments 2: Similarly to Reviewer 3 Comment 2, I was and still am concerned with the role of various components of the catalyst. The authors believe that O₂ is adsorbed and reacts only on Au. This would imply that MoS_x is fully covered with Au, given that MoS_x on TiO₂ is a good photocatalyst itself. However, the authors' own representation of Au clusters on MoS_x, as illustrated in Figure 2a (understood: this is just an illustration, not a proof!) is rather correct, and O₂ reaction at MoS_x cannot be excluded.

Author Reply 2: We are very grateful for the reviewer's comment. In this work, the TiO₂/MoS_x-Au photocatalyst was synthesized by a facile two-step route, including the initial deposition of MoS_x on the TiO₂ surface and the subsequently selective photodeposition of Au onto the MoS_x surface. Herein, with the addition of HAuCl₄ solution into the TiO₂/MoS_x suspension, AuCl₄⁻ ions are selectively adsorbed onto the MoS_x surface and can be in-situ reduced to produce Au nanoparticles on the MoS_x surface under light irradiation. Obviously, it is quite clear that the MoS_x surface cannot be fully covered by Au nanoparticles in the resulting TiO₂/MoS_x-Au photocatalyst.

Photocatalytic experiments (Fig. 3a) indicated that the TiO₂/Au exhibited a H₂O₂-production rate of 24.22 mmol g⁻¹ h⁻¹, which is clearly higher than that of TiO₂/MoS_x photocatalyst (11.95 mmol g⁻¹ h⁻¹), indicating a higher H₂O₂-production efficiency of Au than MoS_x cocatalyst. With the further incorporation of MoS_x into TiO₂/Au, the TiO₂/MoS_x-Au shows a significant enhancement of photocatalytic H₂O₂-production activity (30.44 mmol g⁻¹ h⁻¹), which is 2.55 and 1.3 times higher than that of TiO₂/MoS_x and TiO₂/Au, respectively. The improved H₂O₂-production activity of TiO₂/MoS_x-Au photocatalysts is mainly attributed to enhanced O₂ adsorption on electron-deficient

Au^{δ+} active sites, which can be well confirmed by the in-situ XPS and ΔG_{HOO^*} results. The in-situ XPS (Fig. 6c) indicates that the peaks of Au 4f_{7/2} and Au 4f_{5/2} in the TiO₂/MoS_x-Au are remarkably shifted toward lower binding energies ($\Delta=0.1$ eV) upon light irradiation, suggesting that the photogenerated electrons are directionally transferred from TiO₂ to MoS_x-Au and mainly enriched on the electron-deficient Au^{δ+} sites for promoting photocatalytic H₂O₂-production kinetics. To further verify that the photocatalytic H₂O₂ production is mainly located on Au sites rather than MoS_x sites, the *OOH free energy of MoS_x (ΔG_{HOO^*}) in MoS_x-Au system is calculated using the optimized models (Fig. R1a). It is found that the ΔG_{HOO^*} value on MoS_x is estimated to be -0.65 eV, which is significantly lower than that (4.2 eV) of Au in the MoS_x-Au system (the ideal ΔG_{HOO^*} value for H₂O₂ production is 4.2 eV). Obviously, compared with MoS_x sites, Au cocatalyst in the MoS_x-Au system is more favorable for the photocatalytic H₂O₂ production. Therefore, the O₂-reduction reaction for H₂O₂ production is mainly located on the Au cocatalyst in the present TiO₂/MoS_x-Au photocatalyst.

Our modification to the manuscript: To make Fig. 5f clearer, the O₂ reduction route was provided in Fig. 5f in our revised manuscript.

Fig. R1 (a) The OOH adsorption on MoS_x in the MoS_x-Au and (b) the photocatalytic H₂O₂-production mechanism of TiO₂/MoS_x-Au.

In all, though appreciating this vast multidisciplinary study, I am not convinced that the results, their interpretation, and significance are at the level required for publication in Nature Communications.

We are very grateful for the reviewer's comment. The above comments are carefully considered and presented in our revised manuscript.

Thank you very much again for your kind and appropriate comments. We are sure that these comments can improve the quality of our manuscripts significantly.

Reviewer #3 (Remarks to the Author):

Comments: I was satisfied with the revision.

Author Reply: We are very grateful for the reviewer's comment.

REVIEWERS' COMMENTS

Reviewer #2 (Remarks to the Author):

I am now satisfied with the authors' reply and modifications.